# Lysine acetylome profiling uncovers novel histone deacetylase substrate proteins in *Arabidopsis*

Markus Hartl[1,2,3,§], Magdalena Füßl[1,2,4,§], Paul J Boersema[5,†], Jan-Oliver Jost[6,‡], Katharina Kramer[1], Ahmet Bakirbas[1,4], Julia Sindlinger[6], Magdalena Plöchinger[2], Dario Leister[2], Glen Uhrig[7], Greg BG Moorhead[7], Jürgen Cox[5], Michael E Salvucci[8], Dirk Schwarzer[6], Matthias Mann[5] & Iris Finkemeier[1,2,4,*]

## Abstract

Histone deacetylases have central functions in regulating stress defenses and development in plants. However, the knowledge about the deacetylase functions is largely limited to histones, although these enzymes were found in diverse subcellular compartments. In this study, we determined the proteome-wide signatures of the RPD3/HDA1 class of histone deacetylases in *Arabidopsis*. Relative quantification of the changes in the lysine acetylation levels was determined on a proteome-wide scale after treatment of *Arabidopsis* leaves with deacetylase inhibitors apicidin and trichostatin A. We identified 91 new acetylated candidate proteins other than histones, which are potential substrates of the RPD3/HDA1-like histone deacetylases in *Arabidopsis*, of which at least 30 of these proteins function in nucleic acid binding. Furthermore, our analysis revealed that histone deacetylase 14 (HDA14) is the first organellar-localized RPD3/HDA1 class protein found to reside in the chloroplasts and that the majority of its protein targets have functions in photosynthesis. Finally, the analysis of HDA14 loss-of-function mutants revealed that the activation state of RuBisCO is controlled by lysine acetylation of RuBisCO activase under low-light conditions.

**Keywords** *Arabidopsis*; histone deacetylases; lysine acetylation; photosynthesis; RuBisCO activase

**Subject Categories** Methods & Resources; Plant Biology; Post-translational Modifications, Proteolysis & Proteomics

**Mol Syst Biol. (2017) 13: 949**

## Introduction

Optimal plant growth and development are dependent on fine-regulation of the cellular metabolism in response to environmental conditions (Nunes-Nesi *et al*, 2010). During a day or a season, plants often face rapidly changing environmental conditions such as changes in temperature, light intensity, and water and nutrient availability (Calfapietra *et al*, 2015). Due to their sessile life style, plants cannot escape from environmental perturbations. Instead, plants activate a variety of cellular response mechanisms that allow them to acclimate their metabolism to the environment. Cellular signaling networks are activated within seconds when metabolic homeostasis is perturbed, and these networks regulate the plant's physiology (Dietz, 2015; Mignolet-Spruyt *et al*, 2016). Such signaling networks regulate gene expression, translation, protein activity, and turnover. Post-translational modifications (PTMs) of proteins like phosphorylation, ubiquitination, methylation, and acetylation play a pivotal role in all of these regulatory processes (Hartl & Finkemeier, 2012; Johnova *et al*, 2016). Except for phosphorylation, most of the cellular protein targets and the regulating enzymes of these PTMs are largely unexplored in plants (Huber, 2007). Here, we study the regulation of lysine acetylation.

Lysine acetylation is a post-translational modification (PTM), which was first discovered on histone tails where it is now known to regulate chromatin structure and gene expression (Allfrey *et al*, 1964). The transfer of the acetyl group to lysine neutralizes the positive charge of the amino group, which can affect the biological function of proteins such as enzyme activities, protein–protein, and protein–DNA interactions (Yang & Seto, 2008). Acetyl-CoA serves as

1   Plant Proteomics, Max Planck Institute for Plant Breeding Research, Cologne, Germany
2   Plant Molecular Biology, Department Biology I, Ludwig-Maximilians-University Munich, Martinsried, Germany
3   Mass Spectrometry Facility, Max F. Perutz Laboratories (MFPL), Vienna Biocenter (VBC), University of Vienna, Vienna, Austria
4   Plant Physiology, Institute of Plant Biology and Biotechnology, University of Muenster, Muenster, Germany
5   Proteomics and Signal Transduction, Max-Planck Institute of Biochemistry, Martinsried, Germany
6   Interfaculty Institute of Biochemistry, University of Tübingen, Tübingen, Germany
7   Department of Biological Sciences, University of Calgary, Calgary, AB, Canada
8   US Department of Agriculture, Agricultural Research Service, Arid-Land Agricultural Research Center, Maricopa, AZ, USA
    *Corresponding author. Tel: +49 251 8323805; E-mail: iris.finkemeier@uni-muenster.de
    §These authors contributed equally to this work
    †Present address: Department of Biology, Institute of Biochemistry, ETH Zurich, Zurich, Switzerland
    ‡Present address: Leibniz-Forschungsinstitut für Molekulare Pharmakologie im Forschungsverbund Berlin e.V. (FMP), Berlin, Germany

substrate for lysine acetylation in an enzymatic process catalyzed by different types of lysine acetyltransferases (KATs) (Kleff et al, 1995; Parthun et al, 1996; Yuan & Marmorstein, 2013; Drazic et al, 2016). However, lysine acetylation can also occur non-enzymatically especially at a cellular pH higher than eight (Wagner & Payne, 2013; König et al, 2014a); a level that can be reached during active respiration in the mitochondrial matrix, as well as in the chloroplast stroma during photosynthesis (Hosp et al, 2016). Non-enzymatic acetylation is of particular abundance in bacteria, which additionally contain the highly reactive acetyl-phosphate as metabolite (Weinert et al, 2013). In plants, many organellar proteins from mitochondria and chloroplasts were previously identified as lysine-acetylated (Finkemeier et al, 2011; Wu et al, 2011; König et al, 2014a; Nallamilli et al, 2014; Smith-Hammond et al, 2014; Fang et al, 2015; He et al, 2016; Hosp et al, 2016; Xiong et al, 2016; Zhang et al, 2016; Zhen et al, 2016).

Lysine acetylation can be reversed by lysine deacetylases (KDACs), which were named histone deacetylases (HDA/HDAC) before the more recent discovery of non-histone protein acetylation. KDACs can be grouped into three different families: (i) reduced potassium dependency 3/histone deacetylase 1 (RPD3/HDA1)-like, (ii) HD-tuins (HDT), and (iii) silent information regulator 2 (Sir2) (Pandey et al, 2002; Alinsug et al, 2009; Shen et al, 2015). While the RPD3/HDA1 family has primarily been found in eukaryotes, the Sir2-type deacetylases also occur in bacteria, and the HDT-type deacetylases only occur in plants. The Arabidopsis genome encodes 18 KDACs from the three different families. The largest family comprises the RPD3/HDA1-like with 12 genes, four genes belong to the HDTs and two to the Sir2 family. The RPD3/HDA1 family can be further subdivided into class I (RPD3-like), class II (HDA1-like), and class IV KDACs, of which Arabidopsis possesses 6, 5, and 1 putative members, respectively (Pandey et al, 2002; Alinsug et al, 2009; Shen et al, 2015). Numerous studies have characterized different genes from the Arabidopsis KDAC families over the last two decades (Shen et al, 2015). In particular, HDA6, HDA9, and HDA19 from class I are the most well studied Arabidopsis KDACs and they have been implicated in many important developmental processes such as seed germination, flowering time, as well as plant hormone-related stress responses (Zhou et al, 2005; Benhamed et al, 2006; Chen et al, 2010; Choi et al, 2012; Cigliano et al, 2013; Zheng et al, 2016; Mengel et al, 2017). In terms of protein targets for deacetylation, very little is known about the preferences and targets of the different plant KDACs. While Arabidopsis sirtuin 2 deacetylates selected mitochondrial proteins such as the ATP/ADP carrier (König et al, 2014b), mainly histone H3 and H4 deacetylation has been studied for the other two families of KDACs (Shen et al, 2015).

Here, we report the first comprehensive profiling of putative Arabidopsis KDAC targets by using two different inhibitors of the RPD3/HDA1 family. By this approach, we identify several heretofore-unknown potential targets of the Arabidopsis KDACs in the nucleus and other subcellular localizations including plastids. Additionally, by the use of a peptide-based KDAC-probe, we were able to identify the first KDAC of the RPD3/HDA1 family, which is active in organelles and regulates the activity and activation state of ribulose-1,5-bisphosphate-carboxylase/oxygenase, the key enzyme in photosynthetic $CO_2$ fixation, and the most abundant protein on earth.

# Results

## The Arabidopsis leaf lysine acetylome 2.0

The first two lysine acetylomes of Arabidopsis leaves were reported in 2011, with only around 100 lysine acetylation sites identified (Finkemeier et al, 2011; Wu et al, 2011). Tremendous advances in mass spectrometry, improvements in antibody reagents, and the optimization of the overall protocol now allows a more in-depth profiling of the Arabidopsis lysine acetylome. To be able to quantify acetylome changes upon KDAC inhibitor treatment, we applied an isotopic dimethyl-labeling approach to differentially label two different protein samples (e.g., treatment and control), combined with an enrichment strategy for lysine-acetylated peptides (Fig 1A). For this procedure, proteins extracted from leaves were processed and trypsin-digested via filter-assisted sample preparation. Peptides were isotopically dimethyl-labeled, and samples for comparison were pooled. For proteome quantifications, samples were collected at this step and the rest of the sample was further processed by hydrophilic interaction liquid chromatography fractionation to reduce the peptide complexity. Six to seven fractions were collected and used for immuno-affinity enrichment using anti-acetyllysine agarose beads. Peptides were further processed for high-resolution mass spectrometry, and MaxQuant was used for the data analysis.

Altogether the datasets presented here comprise 2,152 lysine acetylation sites (localization probability > 0.75) on 1,022 protein groups (6,672 identified protein groups in total, Table 1, Datasets EV1–EV5)—this corresponds to 959 novel acetylated proteins and 2,057 novel acetylation sites when compared to the previously published datasets for Arabidopsis (Finkemeier et al, 2011; Wu et al, 2011; König et al, 2014a). A MapMan functional annotation analysis (Thimm et al, 2004) was used for the classification of the lysine-acetylated proteins, applying the TAIR mapping and selecting all identified proteins of the proteome analysis as background population. From the different cellular processes, the functional categories photosynthesis, tetrapyrrole synthesis, gluconeogenesis, redox, TCA cycle, as well as DNA and RNA regulation of transcription were identified as overrepresented as determined by a Fisher's exact test, while processes such as hormone metabolism, cell wall, and secondary metabolism were underrepresented (Fig 1B, at 5% FDR and a 1.5-fold enrichment/depletion cut-off). Based on the classification of localization of proteins using SUBA consensus (Heazlewood et al, 2007), proteins from plastids and nucleus were clearly overrepresented, while proteins from endoplasmic reticulum, vacuole, mitochondrion, plasma membrane, and extracellular space were significantly underrepresented in our dataset (Fig 1C).

Additionally, we analyzed the local sequence context around the acetylation sites using iceLogo (Maddelein et al, 2015) in combination with the Arabidopsis TAIR10 database with all identified proteins as background reference (Fig 1D). Overall, negatively charged amino acids, such as glutamate and aspartate, were significantly enriched in the −1, −2, −3 as well as +1 positions surrounding the lysine acetylation site. In more distant positions, lysine residues were the most strongly enriched on either side of the lysine acetylation site. The sequence motif surrounding the lysine acetylation site appeared different depending on the subcellular localization of the respective proteins. For example, the negatively charged amino acids were more prominent on cytosolic and plastidial

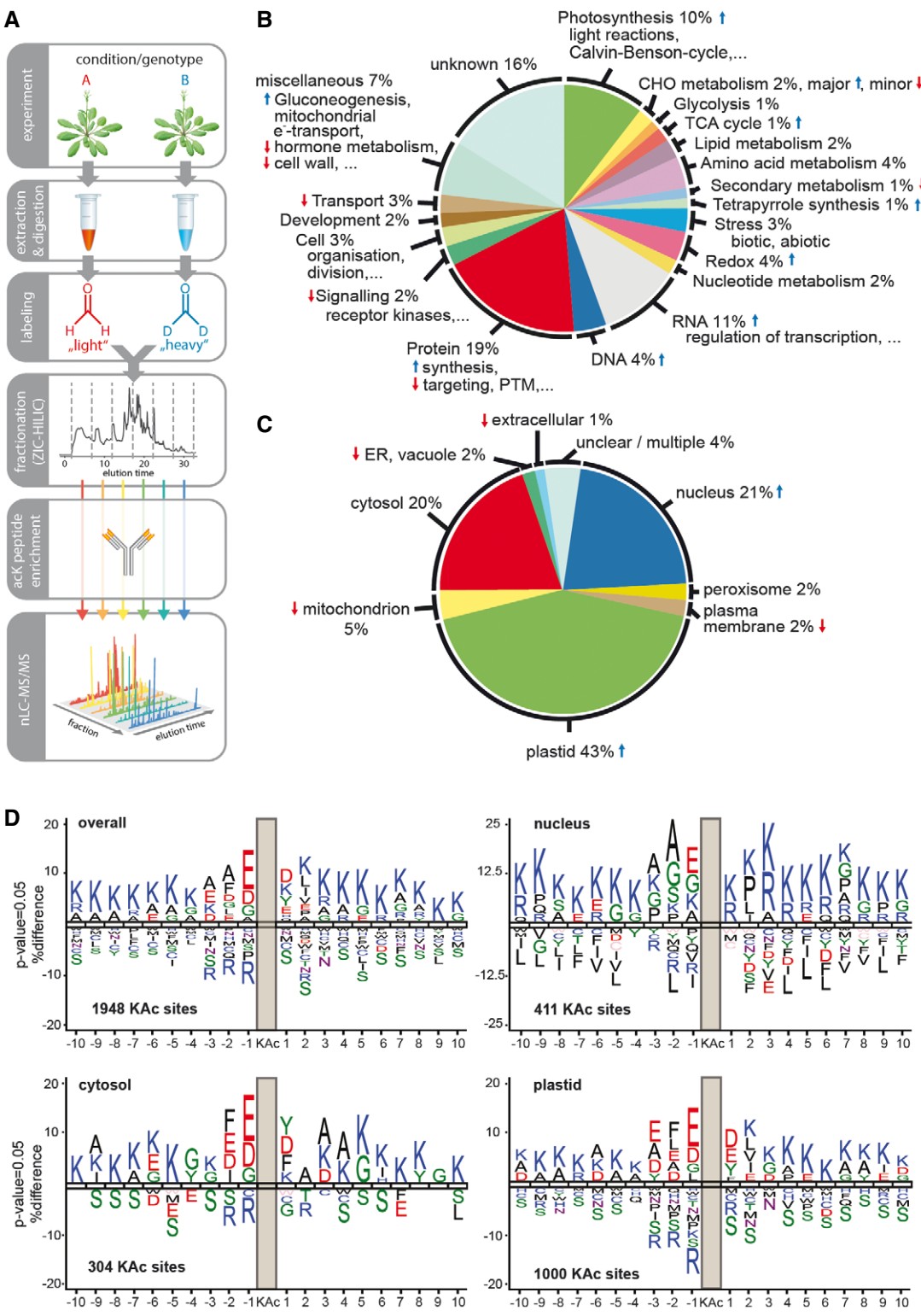

**Figure 1.  Proteome-wide identification and classification of the *Arabidopsis thaliana* lysine acetylome.**

A       Experimental overview.

B, C    Functional classification and subcellular localization of identified lysine-acetylated proteins. Lysine-acetylated proteins identified over all experiments were classified according to MapMan categories and SUBA4 localization information, respectively. Over- or underrepresentation of categories was determined using a Fisher's exact test with all proteins identified at 1% FDR as background population. Blue and red arrows mark categories significantly enriched at 5% FDR (Benjamini–Hochberg) and a 1.5-fold-change cut-off.

D       Sequence logos for all lysine acetylation histone sites with all proteins identified as background population (sequence logos were generated using iceLogo, Maddelein *et al*, 2015).

**Table 1.   Summary of identified features.**

| | | Whole proteome analysis | | Acetyllysine-containing | | |
|---|---|---|---|---|---|---|
| Experiment | Description | Protein groups | Peptides | Protein groups | Peptides | Sites |
| 1 | Apicidin versus Ctrl | 2,384 | 11,188 | 538 | 1,064 | 1,041 |
| 2 | TSA versus Ctrl | 5,107 | 32,809 | 493 | 1,002 | 930 |
| 3 | hda14 versus WT | 2,889 | 13,755 | 545 | 1,133 | 920 |
| 4 | hda14 versus WT low-light | 4,138 | 27,835 | 367 | 756 | 700 |
| 5 | hda14 versus WT thylakoids | 2,904 | 15,064 | 237 | 592 | 546 |
| Total | | 6,672 | 47,338 | 1,022 | 2,405 | 2,152 |

Filters applied: 1% FDR at PSM and protein level, score for modified peptides $\geq$ 35, delta score for modified peptides $\geq$ 6, acetyllysine site localization probability $\geq$ 0.75; contaminants removed.

proteins in comparison with nuclear proteins, as well as the presence of a phenylalanine at position −2. Tyrosine at position +1 was found on cytosolic and plastid proteins, while phenylalanine at position +1 was only found enriched on cytosolic proteins. Interestingly, on the nuclear sequence motifs only positively charged amino acids were found at position +1 as well as generally more neutral amino acids such as glycine and alanine at positions −1 to −3, which are dominating on histone sequence motifs (Fig 1D).

Since 43% of all identified lysine-acetylated proteins are putative plastid proteins, we further analyzed the distribution of those proteins and number of acetylation sites in photosynthesis (Fig 2). About 24% of the proteins from the photosynthetic light reactions were acetylated on at least four lysine residues (Fig 2A). Proteins from the light harvesting complexes (LHC) of both PSII and PSI were heavily acetylated with 29 lysine acetylation sites on LHCII and 16 on LHCI proteins (Fig 2A). All enzyme complexes involved in the carbon fixing reactions (Calvin–Benson cycle) as well as RuBisCO activase (RCA) contained four or more lysine acetylation sites. With 18 lysine acetylation sites, the large subunit of RuBisCO was the most heavily modified of all the proteins (Fig 2B).

### Identification of novel lysine acetylation sites targeted by *Arabidopsis* RPD3/HDA1-type KDACs

Different types of lysine deacetylase inhibitors have been developed in the past decade, which are widely used to modulate the activities of human KDACs in diseases (Newkirk *et al*, 2009). Here, we selected two commonly used KDAC inhibitors, apicidin and trichostatin A (TSA), to target the RPD3/HDA1-type family of KDACs and to profile their potential protein substrates. While apicidin was shown to specifically inhibit class I KDACs, TSA was described as a general inhibitor of class I and class II KDACs in HeLa cells (Scholz *et al*, 2015). For inhibitor treatment, *Arabidopsis* leaf strips were infiltrated either with a mock control or with 5 μM apicidin and 5 μM trichostatin A (TSA), respectively. Experiments were performed in three independent biological replicates, and the leaf strips were incubated for 4 h in the light before harvest. The protein intensities of the biological replicates had a Pearson correlation coefficient of > 0.87–0.98 (Appendix Fig S1), which indicates the robustness of the approach. Site-specific acetylation changes were quantified (Fig 3A and B) in addition to changes on total proteome level as control (Fig 3C and D). No significant changes in the regulation of protein abundances were observed after the inhibitor

treatments, which covered about 67–88% of proteins carrying the identified acetylated sites (Appendix Fig S2). However, the whole proteome analysis did not cover very low abundant proteins without enrichment. Therefore, we cannot exclude that the other sites, for which we were not able to quantify protein ratios, were not regulated due to bona fide stoichiometry differences from inhibited KDAC activity. However, we restricted inhibitor treatment to 4-h incubation time in order to minimize potential changes in protein abundances that might result from KDAC-dependent alterations in gene expression.

For apicidin treatment, 832 lysine acetylation sites were quantified, of which 148 were significantly regulated according to a LIMMA statistical analysis with a FDR cut-off < 5% (Fig 3A; Dataset EV1). As expected for a KDAC inhibitor treatment, most of the lysine acetylation sites (136 in total) were up-regulated (log2-FC 0.4–7.4) after apicidin treatment. The 12 down-regulated lysine acetylation sites comprise mainly multiply acetylated peptides for which peptide variants of lower acetylation status show a down-regulation of particular sites, whereas the corresponding peptide with higher acetylation status shows up-regulation in comparison. Interestingly, while the overall 832 lysine acetylation sites were detected on proteins from various subcellular compartments, 139 of the regulated lysine acetylation sites were found on proteins exclusively localized to the nucleus, such as histones, HATs, proteins involved in the regulation of transcription and signaling (G-protein and light signaling), DNA-repair and cell cycle, as revealed from a SUBAcon analysis (Dataset EV1). Three up-regulated lysine acetylation sites were detected on plastidial proteins including proteins involved in the light reactions (K99, PSAH-1; PSAH-2 log2-FC 0.43) as well as in the Calvin–Benson cycle (K305, SBPase, log2-FC 0.61). Looking at a less stringent *P*-value cut-off < 0.05 (Fig 1B), 182 lysine acetylation sites were found up-regulated of which 29 were found on organellar proteins. While most of these 29 lysine acetylation sites occur on proteins from the plastids, they only show a rather small increase in acetylation level (log2-FC 0.2–0.8) (Dataset EV1).

After TSA treatment, only 37 sites of the 385 quantified lysine acetylation sites were significantly up-regulated with an FDR < 5% (log2-FC 1.4–6.1) (Fig 3B, Dataset EV2). This low number of regulated sites, compared to apicidin treatment, was mainly due to a higher variability in the biological replicates of the TSA treatment. Of the 37 up-regulated lysine acetylation sites, only one was detected on a protein with a plastidial localization (K165,

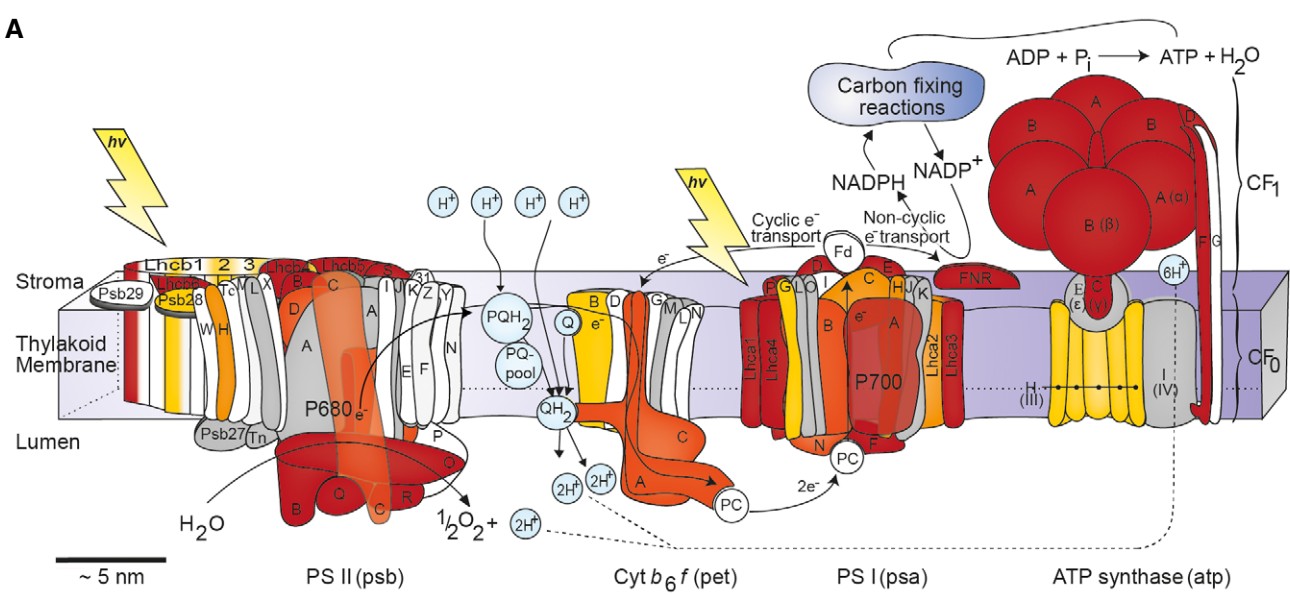

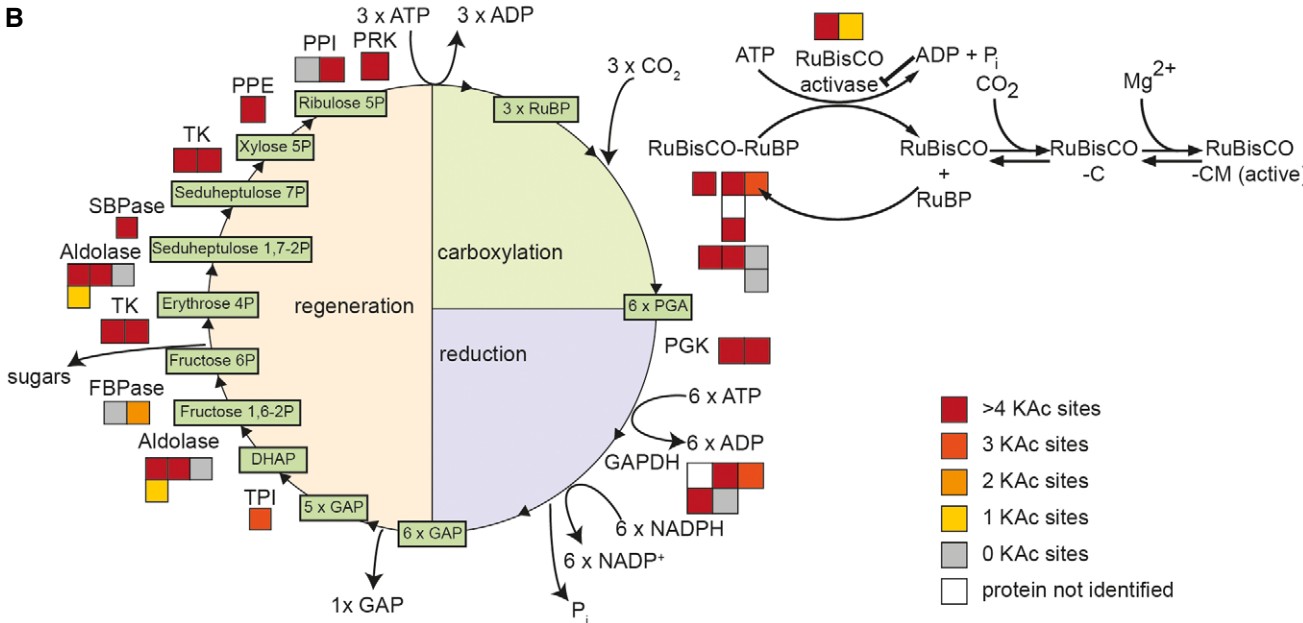

**Figure 2.  Overview of lysine-acetylated proteins in the light reactions (A) and the Calvin–Benson cycle (B) identified in this study in *Arabidopsis*.**

A, B   The classification of proteins into functional bins was performed using MapMan (Thimm *et al*, 2004). Color code: proteins not identified in the LC-MS/MS analyses (white), proteins without identified lysine-acetylated sites (gray), and proteins with one (yellow), two (orange), three (dark orange), or four or more acetylation sites (red). For the Calvin–Benson cycle, each box indicates a separate *Arabidopsis* AGI identifier as indicated in Dataset EV6. Cytb$_6$f, cytochrome b$_6$f; FBPase, fructose-1,6-bisphosphatase; GAPDH, glyceraldehyde-3-phosphate dehydrogenase; PGK, glyceraldehyde-3-phosphate dehydrogenase; PGK, phosphoglycerate kinase; PPE, phosphopentose epimerase; PPI, phosphopentose isomerase; PRK, phosphoribulokinase; PSII, photosystem II; PSI, photosystem I; RuBisCO, ribulose-1,5-bisphosphate-carboxylase/oxygenase; SBPAse, seduheptulose-1,7-bisphosphatase; TPI, triose phosphate isomerase; TK, transketolase. A template of the light-reaction schematic was kindly provided by Jon Nield and modified.

At3g17930, log2-FC 3.6). Analyzing the data with a less stringent *P*-value cut-off ($P < 0.05$) resulted in 72 up-regulated lysine acetylation sites with an average log2-FC of 0.8–6. Among those were two more plastidial proteins, the RCA β1-isoform (K438, log2-FC 1.37) and PSAD-1/2 (K187/K191, log2-FC 1.77). Among the common nuclear targets of apicidin and TSA were several histone proteins. In

total, 19 regulated lysine acetylation sites were found on histone 2A (H2A) and histone 2B (H2B) proteins. On HTB1 two different lysine acetylation sites were specifically and strongly (log$_2$FC at least 2) up-regulated either upon apicidin (K39, K40) or TSA (K28, K33) inhibition (Appendix Fig S3, Dataset EV1A, Dataset EV2A). The same was true for other histones of the H2B family (HTB9, HTB2),

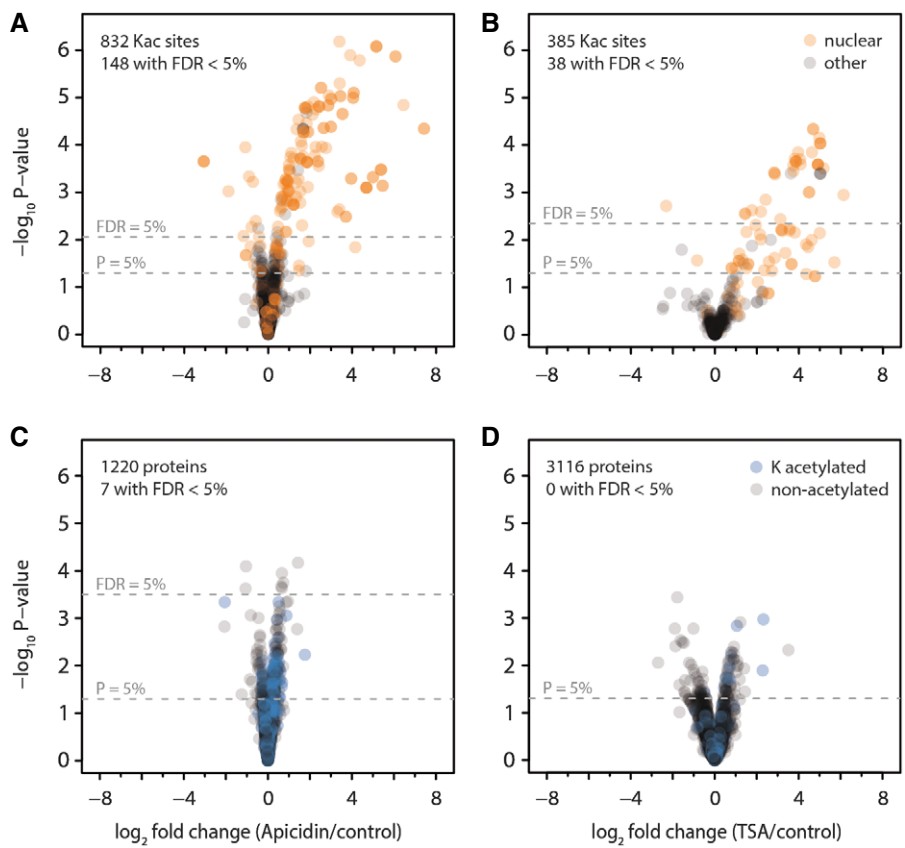

**Figure 3. Differential lysine acetylation and protein expression in *Arabidopsis* leaves after inhibitor treatment.**

A–D   Vacuum infiltration of leaf strips with solutions containing either of the two deacetylase inhibitors apicidin (A, C) or trichostatin A (B, D) versus a buffer control for 4 h leads to differential accumulation of lysine acetylation sites. Volcano plots depict lysine acetylation site ratios (A, B) or protein ratios (C, D) for inhibitor treatment versus control, with *P*-values determined using the LIMMA package. Orange, protein with nuclear localization according to SUBA4 database. Blue, proteins with lysine acetylation sites identified. Dashed lines indicate significance thresholds of either uncorrected *P*-values < 5% or Benjamini–Hochberg corrected FDR < 5%. A missing line indicates that the significance threshold was not reached by any of the data points.

which also showed unique up-regulated sites depending on the inhibitor used. Interestingly, histones of the H2A family showed the same up-regulated lysine acetylation sites upon apicidin and TSA treatment. Overall, between apicidin and TSA treatment, there was an overlap of 25 protein groups (*P*-value < 0.05), which showed enhanced lysine acetylation sites after both treatments (Dataset EV1A, Dataset EV2A). The KDAC inhibitor study revealed that most of the RPD3/HDA1 classes KDACs of *Arabidopsis* have their potential substrate proteins in the nucleus, but that some members also seem to have their targets in other subcellular compartments, such as the plastids.

## HDA14 is the first member of a RPD3/HDA1-family protein to be localized in organelles

Members of the RPD3/HDA1 family are usually localized in the nucleus and/or in the cytosol. Here, we had clear indications that proteins targeted to the chloroplast were found hyper-acetylated upon inhibitor treatment. However, it was not clear whether the hyper-acetylation already occurred due to KDAC inhibition in the cytosol during transit of the proteins to the plastid or whether there exists a plastid-localized member of the RPD3/HDA1-class. Since

KDACs are low abundant proteins, they are usually not detected in leaf proteomes and therefore need to be enriched before detection. Here, we used a recently developed peptide-based KDAC-probe, mini-AsuHd (Dose *et al*, 2016), to pull-down active RPD3/HDA1-class KDACs from leaf extracts and isolated chloroplasts, respectively, in comparison with the mini-Lys probe as background control (Table 2). The mini-AsuHd probe contains a hydroxamate moiety spaced with five carbon atoms to the peptide backbone, which chelates the catalytic $Zn^{2+}$ ion of RPD3/HDA1 family-HDACs with nanomolar affinities (Dose *et al*, 2016). The mini-Lys probe contains a lysine residue instead. Three different *Arabidopsis* KDACs were identified in total leaf extracts (HDA5, 14, 15), while in isolated chloroplasts, only HDA14 was identified (Table 2). Although HDA14 contains a predicted target sequence for plastids, it was reported to be localized in the cytosol in a previous study (Tran *et al*, 2012). To confirm the plastid localization of HDA14, we fused GFP to the C-terminus of the HDA14 protein, instead of the N-terminus as in the previous study. Protoplasts of the stable transformed 35S:HDA14:GFP plants showed that the signal of the HDA14: GFP fusion protein was overlapping with the autofluorescence of the chlorophyll, as well as with a TMRM signal which visualizes

**Table 2.  KDAC pull-down with mini-AsuHd probe.**

| Majority protein IDs | Name | Peptides | MS/MS count | Log2-LFQ CP AsuHd | Log2-LFQ CP Lys | Log2 enrichment CP | Log2-LFQ LF AsuHd | Log2-LFQ LF Lys | Log2 enrichment LF |
|---|---|---|---|---|---|---|---|---|---|
| AT5G61060.1/.2 | HDA5 | 6 | 11 | n.d. | n.d. | n.d. | 24.50 ± 0.12 | n.d. | > 7[a] |
| AT4G33470.1 | HDA14 | 7 | 22 | 24.74 ± 1.83 | 19.54 ± 0.05 | 5.2 | 26.20 ± 0.2 | 21.12 ± 0.49 | 5.1 |
| AT3G18520.1/2 | HDA15 | 2 | 2 | n.d. | n.d. | n.d. | 20.84 ± 0.08 | n.d. | > 3[a] |
| ATCG00490.1 | RBCL | 28 | 545 | 33.65 ± 0.36 | 33.73 ± 0.13 | −0.1 | 33.91 ± 0.09 | 33.75 ± 0.03 | 0.2 |

Selected proteins identified and quantified in pull-downs by LC-MS/MS analysis. Protein abundances are expressed as label free quantification (LFQ) values. Numbers indicate mean log2-transformed LFQ values from two biological replicates of *Arabidopsis* leaves (LF) and isolated chloroplasts (CP). Mini-Lys probes were used as pull-down controls to calculate relative enrichments of proteins. LFQ values for RuBisCO are indicated in all samples as background control.
[a]Estimated enrichment factor assuming a minimum Log2-LFQ threshold of 17.

mitochondria (Fig 4A). Hence, these results indicate a dual localization of HDA14 in mitochondria and chloroplasts. To further confirm the results, we performed a Western blot analysis with HDA14 antiserum on proteins from isolated chloroplasts and mitochondria from WT and stably expressing 35S:HDA14:GFP seedlings and detected the endogenous HDA14 as well as the HDA14-GFP fusion protein in the chloroplast stroma as well as in mitochondria (Appendix Fig S4).

### HDA14 is a functional lysine deacetylase and is mainly inhibited by TSA *in vitro*

We produced a recombinant N-terminally His-tagged HDA14 protein, which lacks the first 45 amino acids of the predicted N-terminal signal peptide (Appendix Fig S5), to investigate the predicted KDAC activity of HDA14. The activity of the purified protein was tested in a colorimetric assay based on the deacetylation of a synthetic acetylated p53 peptide coupled to a chromophore (Dose *et al*, 2012). Using this assay, a deacetylase rate of 0.05/s (± 0.0032) was calculated for His-HDA14 at 100 μM substrate, which is active with both $Zn^{2+}$ or $Co^{2+}$ as cofactors (Fig 4B). Recent publications have shown that recombinant human HDAC8 is more active when the catalytic $Zn^{2+}$ is replaced by $Co^{2+}$ (Gantt *et al*, 2006). However, this is not the case for HDA14, but the enzyme is also active with $Co^{2+}$. Interestingly, apicidin acted only as a weak inhibitor for HDA14 even at concentrations of 100 μM. In contrast, TSA inhibited its activity by 80% at a concentration of 5 μM, the same concentration used in the leaf strip inhibitor experiments.

### HDA14 regulates lysine acetylation levels of plastid proteins related to photosynthesis

To analyze the *in vivo* function of HDA14, a knock-out line (*hda14*) was obtained (Appendix Fig S6) and changes in lysine acetylation site and protein abundances between *hda14* and WT leaves were compared (Fig 5A–F). In total, 832 lysine acetylation sites were identified and quantified from leaves under normal light conditions (Fig 5A; Dataset EV3), and a further 425 lysine acetylation sites were identified from isolated thylakoids (Fig 5B, Dataset EV4), presumably associated with photosynthetic membrane proteins. While no major changes in protein abundances and plant growth were detected for *hda14* in comparison with WT (Fig 5D–F, Appendix Fig S6), 26 lysine acetylation sites on 26 protein groups

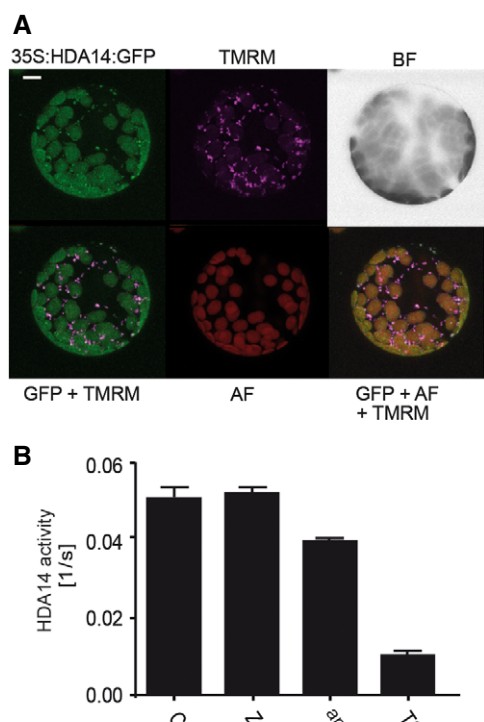

**A**

35S:HDA14:GFP     TMRM     BF

GFP + TMRM     AF     GFP + AF + TMRM

**B**

**Figure 4.  HDA14 protein localizes to the chloroplasts and mitochondria in *Arabidopsis*, and its activity is dependent on cofactors and can be inhibited by deacetylase inhibitors.**

A   GFP localization (green) of the HDA14-GFP fusion constructs in *Arabidopsis* protoplasts (35S:HDA14:GFP) from stable transformants. The mitochondrial marker TMRM is depicted in purple. GFP+TMRM shows the overlay image of 35S:HDA14:GFP and TMRM, AF indicates the chlorophyll autofluorescence and BF the bright-field image of the protoplast. GFP+AF+TMRM represents the overlay image of the three fluorescence channels. Scale bar: 10 μm.
B   Deacetylase activity of the recombinant 6xHis-HDA14 protein using a colorimetric assay. $Co^{2+}$ and $Zn^{2+}$ were used as cofactors, apicidin (100 μM) and trichostatin A (5 μM) as deacetylase inhibitors (n = 5, ± SD).

were increased in between twofold and 80-fold in abundance in the mutant with a FDR < 5%. All of these lysine acetylation sites were detected on proteins localized in the plastid. Another 137 lysine acetylation sites from 122 protein groups were found significantly up-regulated in the mutant but with a lower confidence level

($P < 0.05$). Of these 137 up-regulated lysine acetylation sites, 35 were uniquely identified in the thylakoid fraction and 13 sites were detected in both pull-downs. More than 90% of these proteins are annotated as plastid-localized and are involved in several biochemical processes according to a MapMan analysis (Thimm *et al*, 2004). While around 30% of the proteins have unknown functions, 24% are involved in photosynthesis, 12% in protein synthesis, degradation, and assembly, and around 5% each in lipid metabolism, redox regulation, regulation of transcription, and tetrapyrrole synthesis, as well as 1–3% each are involved in nucleotide metabolism, cell division, ABC transport, secondary metabolism, signaling, organic acid transformation, and amino acid metabolism. Eight of the HDA14 potential target proteins are encoded in the plastome, which further indicates that the deacetylation reaction is occurring within the chloroplast stroma. Among the eight plastome-encoded proteins affected in their acetylation status by the absence of HDA14, the alpha and beta-subunit of the ATP-synthase as well as several photosystem proteins, including the PSII reaction center protein D

and the PSI PsaA/PsaB protein, were identified. These results provide a further indication that HDA14 has a regulatory role in photosynthesis.

The regulation of photosynthesis by post-translational modifications such as phosphorylation and redox regulation is known to be of major importance at low light intensities, for example, during dawn and sunset, when the Calvin–Benson cycle becomes gradually activated or inactivated, respectively, due to changes in stromal pH, ATP, and NADPH levels (Carmo-Silva & Salvucci, 2013; Buchanan, 2016). Hence, we analyzed the acetylation status of the *hda14* plants in comparison with WT after the plants were transferred from normal light (100 μmol quanta m$^2$/s) to low light (20 μmol quanta m$^2$/s) intensities for 2 h. Under these conditions, 36 lysine acetylation sites on 32 protein groups showed a significant increase ($P < 0.05$, 2 to 100-fold), while the total protein abundances of these proteins were unchanged (Fig 5C and F, Dataset EV5). Twenty-six of these proteins are predicted to be localized in plastids. The MapMan analysis revealed that the biological process

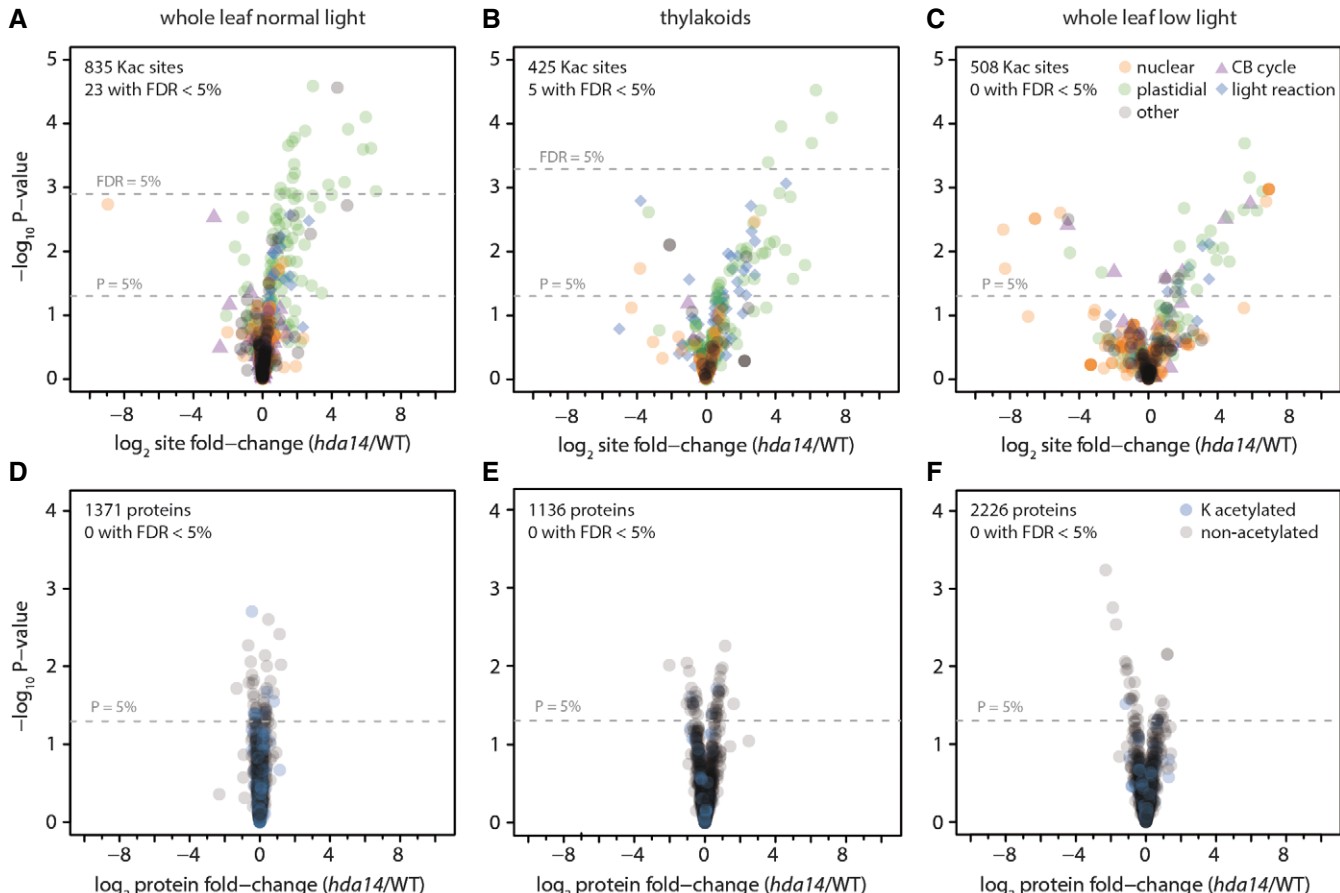

**Figure 5.   Differential lysine acetylation and protein expression in *hda14* versus wild-type leaves under normal light (A, D), in isolated thylakoids (B, E), and under low-light conditions (C, F).**

A–F   Volcano plots depict lysine acetylation site ratios (A–C, top row) or protein ratios (D–F, bottom row) for mutant versus control, with *P*-values determined using the LIMMA package. Orange, protein with nuclear localization; green, protein with plastidial localization; purple triangles, proteins of the Calvin–Benson (CB) cycle; blue diamonds (top row), proteins of the light reaction; localization information according to SUBA4 database. Blue circles (bottom row), proteins with lysine acetylation sites identified. Dashed lines indicate significance thresholds of either uncorrected *P*-values < 5% or Benjamini–Hochberg corrected FDR < 5%. A missing FDR line indicates that the 5% threshold was not reached by any of the data points.

photosynthesis is significantly enriched among those regulated proteins, including RCA (K368/K438, log2-FC 5.89/4.85) as a master regulator of Calvin–Benson cycle activity and RuBisCO large subunit (K474, log2-FC 1.96) itself. RuBisCO catalyzes the carboxylation or alternatively oxygenation of ribulose-1,5-bisphosphate as the first step in either the Calvin–Benson cycle or photorespiration, and thereby enables the photoautotrophic lifestyle of plants. The RuBisCO enzyme is activated by carbamylation of the active site, a process that is dependent on pH, $Mg^{2+}$-ions and requires the removal of sugar-phosphate inhibitors that otherwise block the active site (Portis *et al*, 2008). The removal of these inhibitors requires specific conformational changes to RuBisCO that are induced by RCA, a $AAA^+$-ATPase enzyme. RCA is composed of redox-active alpha isoforms as well as redox-inactive beta-isoforms in *Arabidopsis* (Carmo-Silva & Salvucci, 2013). The RCA activity itself is inhibited under low light by rising ADP concentrations and remains inactive until the photosynthetic electron transport chain again raises the ATP/ADP ratio in response to higher irradiance. The RuBisCO activation state or initial activity, that is, the percentage of active sites free to perform catalysis, as well as total potential activity can be measured by rapid leaf protein extractions (Carmo-Silva *et al*, 2012). Hence, we determined the RuBisCO activity as well as the RuBisCO activation state of the *hda14* plants compared to WT. The results clearly demonstrate that the RuBisCO initial as well as total activity is significantly increased in the *hda14* mutant compared to WT (Fig 6A). While the total activity was increased on average by around 30%, the initial activity was more than doubled in *hda14* compared to WT (Fig 6A), leading to a significantly 90% increased RuBisCO activation state in the mutant under low light (Fig 6B).

Since the lysine acetylation site K438 of the RCA β1-isoform was also found increased after TSA treatment (but not K368), we performed a site-directed mutagenesis on this site in a N-terminally His-tagged RCA-β1 protein. Lysine 438 was exchanged to glutamine (K438Q) and arginine (K438R) to mimic and abolish the lysine acetylation status, respectively. The ATPase activities of the purified mutant RCA proteins were compared to the unmodified WT-like RCA-β1 protein (Fig 6C). Strikingly, the total activity of the K438Q mutant was not affected by this mutation, while the replacement of lysine to arginine led to a strongly diminished enzyme activity. Under low-light conditions, the increase in plastid ADP level plays an important role in the regulation of the RCA activity. Hence, we tested the level of ADP inhibition on the three RCA variants. While the WT-like isoform was inhibited by nearly 19% at an ATP:ADP ratio of 0.11, the activity of the K438Q mutant was only inhibited by about 8%. The K438R mutant, which mimics the non-acetylated state, showed an even stronger ADP inhibition of about 30% under these conditions (Fig 6C). Taken together, the results from this experiment further support that lysine acetylation at K438 leads to a higher RCA and thus RuBisCO activity under low light as observed in the *hda14* mutant.

## Discussion

KDACs have important functions in plant development and acclimation of plants to environmental stresses (Shen *et al*, 2015). So far, these enzymes have mainly been studied with respect to their deacetylase function on histones in plants, despite the large number of different types of lysine-acetylated proteins detected in

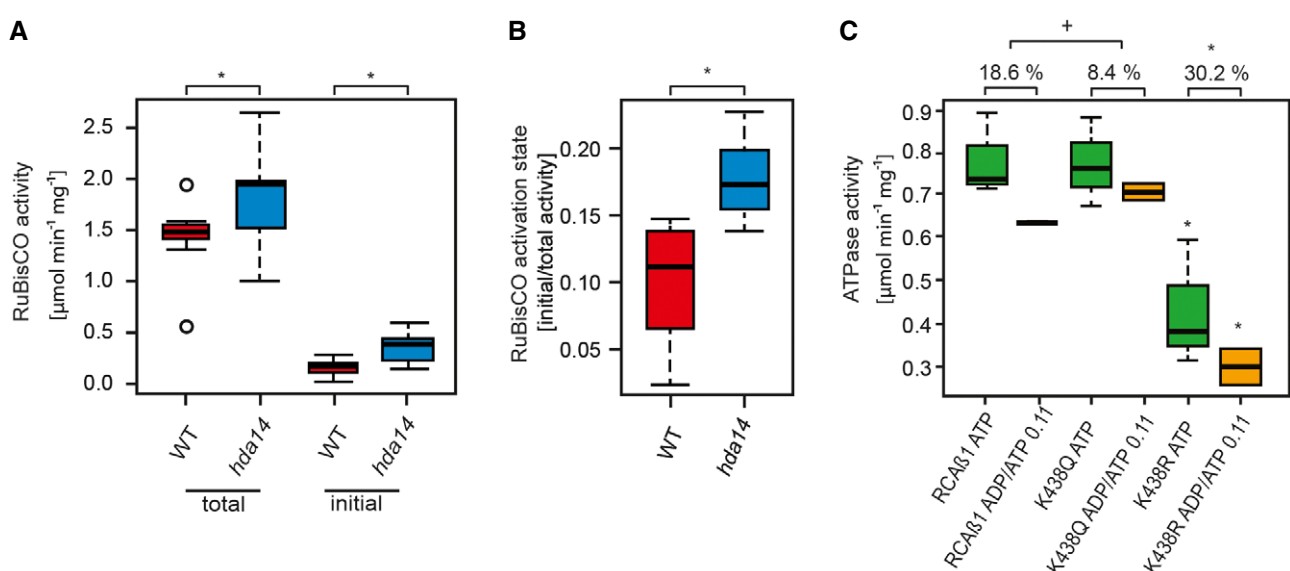

**Figure 6.  RuBisCO activity and RuBisCO activation state are increased in the *hda14* mutant under low-light conditions.**

A   RuBisCO initial and total activity in WT and hda14 in low-light-treated plants. Initial activity was measured directly upon extraction. For the total activity, samples were incubated with $H_2CO_3$ for 3 min to fully carbamylate the active site of RuBisCO ($n$ = 10, *$P$ < 0.05, $t$-test).

B   RuBisCO activation state ($P$ < 0.05, $t$-test).

C   ATPase activity of recombinant 6x-HisRCAβ1 WT, K438Q and K438R with ATP and ADP/ATP = 0.11, respectively ($n$ = 3, *$P$ < 0.05, +$P$ < 0.1, $t$-test). Percentage values on top indicate percent ADP inhibition.

Data information: Boxes indicate lower and upper quartiles of data and whiskers indicate highest and lowest values. Small circles represent outliers. The bars across boxes indicate median values.

recent years (Hosp *et al*, 2016). In this work, we studied the proteome-wide putative targets of the RPD3/HDA1 class of lysine deacetylases in *Arabidopsis* by relative quantification of the changes in the lysine acetylome after inhibitor treatment of *Arabidopsis* leaves with apicidin and TSA. In total, we detected 2,152 lysine acetylation sites in 4-week-old *Arabidopsis* leaves when combining all experiments included in this study. The lysine acetylation sites were found on 1,022 protein groups from all different subcellular compartments and compartment-specific amino acid motifs surrounding the lysine acetylation sites were detected. Similar to human and *Drosophila* sequence motifs, glutamic acid and glycine can be frequently found at position −1 next to the lysine acetylation site also in *Arabidopsis*, while tyrosine, phenylalanine, but not proline, are also enriched in the *Arabidopsis* motifs at position +1, but with a lower frequency (Choudhary *et al*, 2009; Weinert *et al*, 2011). Generally, the acetylated lysines also occur in lysine-rich regions in *Arabidopsis* similar to those described for human and fly (Weinert *et al*, 2011).

In HeLa cells, apicidin was identified as an inhibitor of mainly the RPD3-like KDACs, while TSA inhibits enzymes from both RPD3/HDA1 classes (Scholz et al, 2015). Although we cannot exclude that apicidin and TSA have different specificities for *Arabidopsis* KDACs compared to humans, we observed that recombinant *Arabidopsis* HDA14, which is a HDA1-like KDAC, is not efficiently inhibited by apicidin but by TSA. This supports the notion that similar specificities of both inhibitors exist for the *Arabidopsis* KDACs as well. In *Arabidopsis*, four KDACs belong to the RPD3-like group, including HDA1/19, HDA6, HDA7, and HDA9 (Hollender & Liu, 2008). Two additional KDAC genes, *HDA10* and *HDA17*, are closely related to *HDA9*, but the predicted proteins lack a catalytic domain and therefore are probably inactive. Lysine acetylation sites on 91 protein groups were significantly ($P < 0.05$) up-regulated after apicidin treatment and are therefore most likely substrates of at least one of these four KDACs. The dominant nuclear localization among these proteins fits to the observed localizations of the RPD3-like KDACs in the nucleus. While HDA1/19, HDA6, and HDA9 were detected mainly in the nucleus, the localization of HDA7 has yet to be determined. Although the predicted HDA7 protein contains both a nuclear localization sequence as well as a nuclear export signal, it is unclear to what extent this protein is active due to its low expression level in most tissues.

From the 91 target protein groups identified upon inhibition with apicidin, only 14 are histone-like proteins. Hence, we identified 77 new candidate protein groups, which are potential substrates of the RPD3-like KDACs in *Arabidopsis*. This list of potential target proteins with the exact information on their acetylation sites can be regarded as valuable resource for future studies on the KDAC functions in plant stress response and development. Interestingly, a high mobility group box protein with ARID/BRIGHT DNA-binding domain (At1g76110) was identified as one of the substrate proteins, which was also regulated upon TSA treatment. These types of proteins have been identified as interaction partners of human HDAC1/2 (Joshi *et al*, 2013). Furthermore, a physical interaction was previously detected between *Arabidopsis* HDA6 and the histone H3.3 (At4g40030) (Earley *et al*, 2006). We identified several peptides of histone H3-like proteins that were up-regulated by more than 2-fold at the positions K9, 14, 18, 23, 27, 36, and 37 after apicidin treatment, but less so after treatment with TSA. Several of

these lysine acetylation sites on histone H3 are of great importance for chromatin regulation and remodeling [e.g., (Mahrez *et al*, 2016)]. For example, H3K9 acetylation was found to be associated with actively transcribed genes and has a strong impact on various developmental processes in plants (e.g., Ausín *et al*, 2004; Benhamed *et al*, 2006). Differences in the strength of TSA and apicidin inhibition could be explained by differences in the uptake of the inhibitors into the *Arabidopsis* cells, as well as by differences in the Ki values of the different *Arabidopsis* KDACs for these chemicals. Furthermore, our data indicate that TSA might not be effectively taken up into plastids, since the recombinant HDA14 protein was strongly inhibited by TSA, but only few plastid proteins were affected by TSA treatment of *Arabidopsis* leaves.

In addition to the many new KDAC inhibitor target proteins in the nucleus, there were also several interesting candidate proteins identified in the cytosol, such as the FRIENDLY protein (At3g52140), which is required for correct distribution of mitochondria within the cell. In a previous study, we already demonstrated that two lysine sites, which can be acetylated, regulate FRIENDLY function (El Zawily *et al*, 2014).

After TSA treatment, lysine acetylation sites from unique protein groups were regulated, which were not affected by apicidin treatment, indicating that those sites are specifically regulated by HDA1-type HDACs. These proteins included RCA-β1, photosystem I subunit D-2, ribosomal L6 family protein, S-adenosyl-L-methionine-dependent methyltransferases superfamily protein, the telomere repeat binding factor 1, and a histone H2B protein (Dataset EV2). Candidates of KDAC proteins from the HDA1-type group that might be responsible for the regulation of the lysine acetylation sites of these proteins include HDA5, 8, 14, 15, and 18, which cluster together with the human class 2 KDACs (Alinsug *et al*, 2009). By using a hydroxamate-based KDAC-probe, which allows the enrichment of active RPD3/HDA1-class KDACs from protein extracts, we were able to detect HDA5, 14, and 15 in total leaf extracts of 4-week-old *Arabidopsis* leaves. By using the same probe, we previously enriched all class 1 and class 2b KDACs from HeLa cells, indicating that the probe is able to bind all types of RPD3/HDA1-class KDACs (Dose *et al*, 2016). Hence, we conclude that HDA5, 14, and 15 are the most abundant KDACs in *Arabidopsis* leaves. Strikingly, both TSA and apicidin treatment resulted in an increased acetylation of plastid proteins involved in photosynthesis. Here, we identified HDA14 as the first organellar-localized RPD3/HDA1 class protein which is active as a KDAC and which has the majority of its candidate target proteins in the plastid stroma. At the concentration of apicidin used in our study, HDA14 was not significantly inhibited in its activity, which further supports the observation that apicidin mainly inhibits RPD3-like KDACs. Hence, the plastid target proteins, which showed mildly increased lysine acetylation after apicidin treatment, might be regulated by an unknown RPD3-like deacetylase. For example, we identified six lysine acetylation sites on the TROL protein, which is required for anchoring the ferredoxin-NADP reductase (FNR) to the thylakoid membranes and to sustain efficient linear electron flow in the light reactions of photosynthesis. While K328 and K348 of TROL were more than twofold increased in their acetylation level in the *hda14* loss-of-function mutant, only K337 showed a 1.2-fold increased acetylation after apicidin treatment. Lysine acetylation sites on FNR itself were not significantly regulated upon KDAC inhibition. Here, we identified four lysine

acetylation sites on both FNR1 (K287, 290, 321, 325) and FNR2 (K90, 243, 299, 330) isoforms, of which only two have been previously reported (Lehtimaki *et al*, 2014).

By analyzing the acetylome of *hda14* mutants, we were able to identify the unique substrate proteins of HDA14. HDA14 was previously identified as a nuclear/cytosolic protein based on enrichment in the microtubule fraction of a red fluorescent-tagged version of the protein (Tran *et al*, 2012). However, the N-terminal location of this tag would hinder the protein from entering the chloroplast. Moreover, the authentic N-terminus of HDA14 contains a clear signal sequence for the plastids as predicted by bioinformatical analysis (Alinsug *et al*, 2009). By using the Asu-Hd probe on isolated plastid fractions as well as by using C-terminal GFP-tagged fusion proteins, we were able to confirm the predicted plastid localization of HDA14. Furthermore, most of the candidate HDA14 substrate proteins identified reside in the plastids and are involved in metabolism and photosynthesis.

With the analysis of the HDA14-dependent acetylome, we found that the RCA-β1 site K438 is a substrate site of HDA14. Increased acetylation of this site reduced the ADP sensitivity of the RCA protein, which plays an important role for the Calvin–Benson cycle activation under low light intensities when the ADP/ATP ratio in plastids is still high (Carmo-Silva & Salvucci, 2013). Since the alpha-isoform of RCA in *Arabidopsis* is considerably more sensitive to inhibition by ADP than the beta-isoform (Carmo-Silva & Salvucci, 2013), the effect of acetylation of K438 of RCA-β1 in relieving ADP inhibition might be mediated by RCA-α through an effect of acetylation on subunit (i.e., alpha–beta) interaction.

Lysine acetylation on *Arabidopsis* RCA was detected in a previous study (Finkemeier *et al*, 2011), but on a different lysine residue and the functional consequences were not studied so far. In addition to increased acetylation, we also observed increased RuBisCO activity under low-light conditions in the *hda14* mutant, which co-occurred with increased acetylation at K474 on the RuBisCO large subunit next to the strongly increased acetylation of K368 and K438 on RCA and on carbonic anhydrase (K269, At3g01500). Acetylation on all of these proteins might play an important role in fine-tuning of RuBisCO activity. In contrast to our results obtained in the *hda14* mutant, two independent previous studies revealed that a decrease in RuBisCO acetylation resulted in a higher activity of the enzyme (Finkemeier *et al*, 2011; Gao *et al*, 2016). However, K474 on the RuBisCO large subunit (RBCL) was not detected in either of these studies, and hence, this site could have a different role than acetylation on any of the other 18 acetylation sites detected here and elsewhere.

In conclusion, in this study, we were able to define the hereto-fore-unknown acetylation candidate target proteins of RPD3/HDA1 class HDACs in *Arabidopsis* and specifically those of HDA14, as the first identified RPD3/HDA1 KDACs in organelles. Furthermore, our study revealed that about 10% of the detected lysine acetylation sites can be regulated by these types of KDACs in *Arabidopsis* leaves. Many sites might be specifically regulated under certain environmental or developmental conditions due to changes in KDAC activities, as we observed for low-light conditions for example. The activity of KDAC themselves might be regulated via post-translational modifications (Mengel *et al*, 2017) or by change in interaction partners that lead to the formation of different KDAC complexes (Dose *et al*, 2016). Future studies, with more detailed

analyses of individual lysine sites in proteins and the analysis of further KDAC mutants and environmental conditions, will allow unraveling this complex network of fine-tuning of protein functions and interactions by lysine acetylation. Since lysine acetylation sites can act as molecular switches, they could be engineered in plant proteins to regulate cell-signaling cascades, the expression of certain genes, or to modulate the activities of metabolic enzymes. Furthermore, due to recent advances in advances in CRISPR/CAS technologies, lysine acetylation sites can be used for site-directed mutagenesis also in crop plants. Modifying these lysine residues to constitute acetylated or non-acetylated mimics ideally will allow a switching of metabolic activities and outputs that have the potential to enhance plant yields or direct metabolism in a way to enhance the accumulation of metabolic intermediates to increase the nutritional values of crops and thereby indirectly promote human health.

# Materials and Methods

### Plant material and growth conditions

Arabidopsis *thaliana* (Col-0) plants were grown for 4 weeks in a climate chamber using a 12-h light/12-h dark (21°C) photoperiod with a light intensity of 100 μmol quanta m$^2$/s and 50% relative humidity. For low-light treatments, plants were transferred to 20 μmol quanta m$^2$/s for 2 h before harvest. For growth on plates, *Arabidopsis* seeds were sterilized and transferred to half-strength Murashige–Skoog medium supplemented with 0.8% phytoagar. The *hda14* line (SALK_144995C) was obtained from the Nottingham *Arabidopsis* stock center (NASC) and PCR screened according to Salk Institute Genomic Analysis Laboratory instructions (O'Malley *et al*, 2007) using the following primers: HDA14_LP 5′-GAAAC ATGTCACGCAAAAATG-3′, HDA14_RP 5′-TTTTGTTGGTTTGCTTC TTCG-3′, and the TDNA primer SALK-Lb1.3 5′-ATTTTGCCGA TTTCGGAAC-3′. PCR products were run on 1% agarose Tris–acetate (TAE: 40 mM Tris, 20 mM acetate, 1 mM EDTA, pH 8.0) and visualized by UV illumination upon ethidium bromide staining.

### Trichostatin A and apicidin treatment

About 20 fully expanded leaves from 4-week-old *Arabidopsis* plants were pooled and cut into 2-mm-diameter leaf slices (for each biological replicate). After vacuum infiltration in effector solutions (three times for 5 min), the leaf slices were incubated at 100 μmol quanta m$^2$/s for 4 h. All solutions used for infiltrations were made in 1 mM MES pH 5.5 (KOH). All chemicals were purchased from Sigma-Aldrich (Gillingham, Dorset, UK). All stock solutions were dissolved in DMSO. Control experiments were then performed with DMSO added in same concentrations without effectors. Leaf material was briefly dried on tissues for harvest and flash-frozen in liquid nitrogen.

### GFP fusion and plant transformation

Entry clones for Gateway cloning were generated with the pENTR/ SD/TOPO vector (Invitrogen™). The open reading frame of *HDA14* (At4g33470) without stop codon from *Arabidopsis* (Col-0) was amplified from cDNA using the following primers: 5′-CACCATGTC CATGGCGCTAATTGT-3′ and 5′-TAAGCAATGAATGCTTTTGGCTC

TC-3′. LR reactions were performed for recombination into the pK7GW2 vector (Karimi *et al*, 2007). The vector construct was verified by sequencing and transformed into *Agrobacterium tumefaciens* strain C58 followed by floral dip transformation of *Arabidopsis* (Col-0) plants (Clough & Bent, 1998). Transformants were selected by germination of seeds on MS-agar plates containing kanamycin (50 µg/ml). Resistant plants were transferred to soil and propagated.

### RNA isolation and RT–PCR

Total RNA of *Arabidopsis* leaves was extracted using Trizol® (Invitrogen™) followed by chloroform extraction, and precipitation with isopropanol and subsequently LiCl₂. The quality and quantity of the RNA were confirmed on agarose gels and a UV-spectrometer. Complementary DNA (cDNA) was synthesized from DNase-treated RNA with SuperScriptIII reverse transcriptase (Invitrogen™) following the manufacturer's instruction and using $dT_{20}$. Real-time qPCR was carried out in triplicate in an iQ™5 Multicolor Real-Time PCR Detection System (Bio-Rad) using iQ™SYBR Green Super Mix (Bio-Rad) and gene-specific primers: HDA14-F 5′-ATCTGTGGCAGACT CGTTTCG-3′, HDA14-R 5′-TCGCACCTTTCTCATTGGTTC-3′. Levels of selected transcripts in each sample were calculated using a standard curve method (Finkemeier *et al*, 2013). Expression levels of the HDA14 transcript were normalized to *ACTIN2* (At3g18780) transcript as housekeeping gene using the following primers: actin2-F 5′-CTGTACGGTAACATTGTGCTCAG-3′ and actin2-R 5′-CCGATCCA GACACTGTACTTCC-3′.

### Protoplast isolation and confocal laser scanning microscopy

Protoplast isolation was performed from 4-week-old *Arabidopsis* leaves after the tape-sandwich method (Wu *et al*, 2009). Staining with 20 nM TMRM (Sigma) was performed according to the manufacturer's protocol. Imaging was performed with a spectral TCS SP5 MP confocal laser scanning microscope (Leica Microsystems, Mannheim, Germany) using an argon and DPSS-Laser laser, respectively, at an excitation wavelength of 488 nm (eGFP) and 543 nm (TMRM). The water immersion objective lens HCX PL APO 20.0× 0.70 IMM UV was used for imaging in multitrack mode with line switching. eGFP fluorescence and TMRM fluorescence were measured at 500–530 and 565–615 nm, respectively.

### Heterologous expression and purification of recombinant HDA14 protein

cDNAs were amplified by PCR excluding the coding region for the 45aa signal peptide using the following primers: HDA14p-F: 5′-TTTAGTACAGAGAAGAATCCTCTATTACCATCT-3′ and 5′-TCAAA CAAATTCACCTTATAAGCAATG-3′. The PCR product was cloned into pEXP-5-NT/TOPO® TA (Invitrogen™), which allows expression and purification of the recombinant, N-terminally 6× His-tagged protein. Vector constructs were verified by sequencing. After transforming *E. coli* BL21(DE3) (Invitrogen™) with the expression vector, the recombinant protein was expressed using the EnPresso system (BioSilta, Germany) as described before (Jost *et al*, 2015): 500 ml of EnPresso medium was mixed with 12.5 µl $ZnCl_2$ (1 mM) solution and 25 µl of the "EnZ I'm" mix. Freshly prepared medium was

inoculated 1:100 with a 6-h-cultivated pre-culture at 37°C under gentle shaking (160 rpm) and incubated overnight. Subsequently, the temperature was reduced to 25°C and a "booster tablet" and 50 µl "EnZ I'm" mix were added to the culture medium followed by 500 µM IPTG. The culture was incubated for 24 h at 250 rpm. The cells were harvested by centrifugation (10 min at $3,000 \times g$), and the pellet was resuspended in PBS buffer (pH 8.0) and lysed with a homogenizer (EmulsiFlex-C5, Avestin) at 4°C. The cleared lysate (centrifugation for 20 min at $30,000 \times g$) was incubated with 1 ml of Ni-NTA Agarose slurry (Qiagen) for 2 h at 4°C. The resin was washed with 50 ml PBS, pH 8.0, 4°C, and the protein was eluted with 300 mM imidazole in HDAC buffer (8 mM KCl, 100 mM NaCl, 10 mM HEPES, pH 8.0). Pure fractions were combined and dialyzed against HDAC buffer containing 10 mM EDTA and subsequently against HDAC buffer supplied with 0.5 mM EDTA. The sample was concentrated with a centrifugation filter device with 10 kD MWCO (Amicon Ultra, Merck Millipore), supplied with 20% (v/v) glycerol and stored at −80°C until usage.

### HDA14 activity assay

The deacetylation assays were performed with a previously described p53-derived peptide substrate containing the chromophore 5-amino-2-nitrobenzoic acid (p53-5,2-ANB) (Dose *et al*, 2012). To produce the apoenzyme, the purified HDA14 protein was first dialyzed against 10 mM EDTA and 1 mM DTT and in a second step against 0.5 mM EDTA to remove bound metal ions. For the enzyme asssay, HDA14 was supplied with either $Zn^{2+}$ or $Co^{2+}$ ions by incubating the enzyme solution with 1 mM of $ZnCl_2$ or $CoCl_2$ on ice for 30 min. Deacetylation assays were performed by incubating 1 µM of either $Zn^{2+}$ or $Co^{2+}$ supplied HDA14 with 100 µM p53-5,2-ANB substrate in HDAC reaction buffer (10 mM HEPES, 100 mM NaCl, 8 mM KCl, 10 µM BSA, pH 8.0) in a total volume of 50 µl at RT. The reaction was stopped after 10 min by adding 10 µl quenching solution (6.25 µM TSA in 0.1% (v/v) TFA) and developed by adding 10 µl of trypsin solution (6 mg/ml). After 30 min of trypsinization, the reaction mixture was supplied with 70 µl of HDAC reaction buffer, transferred into a 100-µl quartz cuvette, and the absorbance was monitored at 405 nm in a photometer (Helma, Germany). Studies with inhibitors apicidin (5 and 100 µM) and TSA (5 µM) were performed by adding these compounds to the assay before the reactions were started. All rates were normalized to the concentration of HDA14.

### Western blot analyses

Proteins were separated on 12% SDS–polyacrylamide gels, blotted onto nitrocellulose membrane, and incubated overnight with the primary antibodies. The secondary IRDye 800CW antibody (LI-COR) was used in a 1:10,000 dilution and detected with the Odyssey reader (LI-COR).

### RuBisCO activity measurements for activation state determination

For determination of the RubisCO activation state under low-light conditions, plants were transferred to low irradiation (20 µmol quanta m²/s) for 5 h at 21°C, then harvested, and frozen in liquid

nitrogen. RuBisCO initial and total activity were assayed by incorporation of $^{14}CO_2$ into acid-stable products (Salvucci, 1992). The leaves were homogenized in extraction medium [100 mM Tricine–NaOH pH 8.0, 1 mM EDTA, 5% polyvinylpyrrolidone (PVP-40), 5% polyethylene glycol 3350 (PEG3350), 5 mM (DTT), and protease inhibitor cocktail (Roche)]. Initial activities were measured immediately upon extraction, whereas total activities were measured after 3-min incubation in assays without RuBP to fully carbamylate the enzyme (Carmo-Silva *et al*, 2012). For each sample, assays were conducted in duplicate. Initial and total activities were used to calculate RuBisCO activation state, that is, (initial/total activity × 100) = % activation.

### Purification and assay of RCA

The coding sequence of the RCAβ1 spliceform (At2g39730.2) was amplified from *Arabidopsis* cDNA using the following primers: 5′-CTCCGATATCTTACTTGCTGGGCTCCTTT-3′ and 5′-TTTTTGATA TCTCAAACCTCTGTTTTACC-3′ introducing SacI and EcoRV restriction sites for cloning into pCDFDuet-1 (Novagen®). Site-directed mutants of RCA K438R and K438Q were introduced with the QuikChange Site-Directed Mutagenesis Kit (Agilent Technologies) using the following primers: RCAβ2-K438R 5′-GAACTTTCTACGGT<u>AGA</u>A CAGAGGAAAAGG-3′ and RCAβ2-K438Q 5′-GAACTTTCTACGGT<u>CAA</u> ACAGAGGAAAAGG-3′. The N-terminally 6-His-tagged protein was expressed and purified from Rosetta-gami cells (Novagen®) as described detail in (Barta *et al*, 2011). ATPase activity of 5 µg recombinant RCA was measured for 1 min at 23°C in 50 µl reaction buffer (100 mM HEPES-KOH (pH 8.0), 20 mM MgCl$_2$) containing 500 µM ATP and 500 µM ATP and 55 µM ADP, respectively. The reaction was heat inactivated at 95°C. The ATP consumption was determined using the KinaseGlo Max Luminescent Assay Kit (Promega) according to the manufacturer's protocol.

### Isolation of intact chloroplasts and mitochondria

Chloroplasts were isolated from dark incubated (12 h) 5-week-old rosette leaves of *Arabidopsis*. Leaves were homogenized in ice-cold HB-buffer (0.45 M sorbitol, 20 mM Tricine-KOH pH 8.45, 10 mM EDTA, 10 mM NaHCO$_3$, 0.1% BSA, and 2 mM sodium ascorbate). Chloroplasts were purified on a Percoll gradient (40–80%) and resuspended in sorbitol buffer (0.3 M sorbitol, 20 mM Tricine-KOH pH 8.45, 2.5 mM EDTA, and 5 mM MgCl$_2$, 2 mM sodium ascorbate). Mitochondria were isolated as described previously (König *et al*, 2014a).

### Isolation of thylakoids

Chloroplasts were lysed in 2 ml TMK buffer (50 mM HEPES/KOH pH 7.5, 0.1 M sorbitol, 5 mM MgCl$_2$, 10 mM NaF), and thylakoid membranes were sedimented at 14,000 × *g*.

### Preparation of cell extracts and enrichment of active histone deacetylases

Leaves from 5-week-old *Arabidopsis* plants were homogenized in extraction buffer (50 mM Tris–KOH (pH 7.5), 150 mM NaCl, 10% [v/v] glycerol, 5 mM dithiothreitol (DTT), 1% [v/v] Triton X-100,

and protease inhibitor cocktail (Sigma-Aldrich). Homogenates were centrifuged at 14,000 × *g*, and protein concentration of the supernatant was determined with the Pierce 660 nm Protein Assay (Thermo Fisher Scientific). All protein extracts were desalted on PD-10 Desalting Columns (GE Life Sciences), and the samples were eluted with immunoprecipitation buffer (50 mM Tris, pH 7.5, 150 mM NaCl, 10% [v/v] glycerol).

The immobilized peptide probes, mini-AsuHd, and mini-Lys (Dose *et al*, 2016) were equilibrated with immunoprecipitation buffer two times and incubated with the protein extracts overnight at 4°C, under constant rotation. The next day, the beads were gently pelleted by centrifugation. The beads were transferred onto micro-centrifugal filter system (Amchro GmbH) and washed five times with 1 ml immunoprecipitation buffer. Proteins bound on beads were subjected to on-bead digestion. Proteins were denatured in 6 M urea prepared in 0.1 M Tris–HCl (pH 8.0), 1 mM CaCl$_2$ and reduced with 5 mM DTT. Reduced cysteines were alkylated with 14 mM chloroacetamide for 30 min. Excess chloroacetamide was quenched with DTT. Proteins were trypsinated at a urea concentration of 1 M and a trypsin (Sigma-Aldrich) to protein ratio of 1:100 at 37°C. Resulting peptides were desalted on SDB-RPS and C18 Stage-Tips, respectively (Rappsilber *et al*, 2007; Kulak *et al*, 2014).

### Protein extraction, peptide dimethyl labeling, and lysine-acetylated peptide enrichment

Frozen leaf material was ground to fine powder in liquid nitrogen and extracted using a modified filter-assisted sample preparation (FASP) protocol with 30k MWCO Amicon filters (Merck Millipore) as described in detail in Lassowskat *et al* (2017). Digested peptides were dimethyl-labeled on C18 Sep-Pak plus short columns (Waters) as described previously (Boersema *et al*, 2009; Lassowskat *et al*, 2017). Equal amounts of light and medium-labeled peptides (3–5 mg) were pooled for each replicate and the solvent evaporated in a vacuum centrifuge. The dried peptides were dissolved in 1 ml TBS buffer (50 mM Tris–HCl, 150 mM NaCl, pH 7.6), and pH was checked and adjusted where required. 15 µg peptide mixture was stored for whole proteome analysis. About 10 mg of the pooled labeled peptides was resuspended in 2 ml 95% solvent A (95% acetonitrile, 5 mM ammonium acetate) and 5% buffer B (5 mM ammonium acetate) and fractionated with a flow rate of 500 µl/min on a Sequant ZIC®-HILIC column (3.5 µm, Merck) using a segmented linear gradient of 0–60%. The fractions were combined to seven final fractions and dried in a vacuum centrifuge. Peptides were resuspended in IP buffer (50 mM Tris–HCl pH 7.6, 150 mM NaCl), and the concentration was determined on the spectrophotometer at 280 nm. Lysine-acetylated peptide enrichment was performed as previously described with 1 mg peptide per fraction (Hartl *et al*, 2015; Lassowskat *et al*, 2017). After enrichment, the eluted peptides were desalted using C18 StageTips and dried in a vacuum centrifuge.

### LC-MS/MS

Dried peptides were redissolved in 2% ACN, 0.1% TFA for analysis. Total proteome samples were adjusted to a final concentration of 0.2 µg/µl. Samples were analyzed using an EASY-nLC 1000 (Thermo Fisher) coupled to a Q Exactive, Q Exactive Plus, and an Orbitrap Elite mass spectrometer (Thermo Fisher), respectively. Peptides

were separated on 16 cm frit-less silica emitters (New Objective, 0.75 μm inner diameter), packed in-house with reversed-phase ReproSil-Pur C18 AQ 3 μm resin (Dr. Maisch). Peptides (5 μl) were loaded on the column and eluted for 120 min using a segmented linear gradient of 0% to 95% solvent B (solvent A 5% ACN, 0.5% FA; solvent B 100% ACN, 0.5% FA) at a flow rate of 250 nl/min. Parameters for the different machines are listed in Dataset EV1.

### MS data analysis

Raw data were processed using MaxQuant software version 1.5.2.8 (http://www.maxquant.org/) (Cox & Mann, 2008). MS/MS spectra were searched with the Andromeda search engine against the TAIR10 database (TAIR10_pep_20101214; ftp://ftp.arabidopsis.org/home/tair/Proteins/TAIR10_protein_lists/). Sequences of 248 common contaminant proteins and decoy sequences were automatically added during the search. Trypsin specificity was required, and a maximum of two (proteome) or four missed cleavages (acetylome) were allowed. Minimal peptide length was set to seven amino acids. Carbamidomethylation of cysteine residues was set as fixed, oxidation of methionine, and protein N-terminal acetylation as variable modifications. Acetylation of lysines was set as variable modification only for the antibody-enriched samples. Light and medium dimethylation of lysines and peptide N-termini were set as labels. Peptide–spectrum matches and proteins were retained if they were below a false discovery rate of 1%, modified peptides were filtered for a score ≥ 35 and a delta score of ≥ 6. Match between runs and requantify options were enabled. Downstream data analysis was performed using Perseus version 1.5.5.3 (Tyanova *et al*, 2016). For proteome and acetylome, reverse hits and contaminants were removed, the site ratios log2-transformed, and flip-label ratios inverted. For quantitative lysine acetylome analyses, sites were filtered for a localization probability of ≥ 0.75. The "expand site table" feature of Perseus was used to allow separate analysis of site ratios for multiply acetylated peptides occurring in different acetylation states. Technical replicates were averaged, and proteins or sites displaying less than two out of three ratios were removed. The resulting matrices for proteome and acetylome, respectively, were exported and significantly differentially abundant protein groups and lysine acetylation sites were determined using the LIMMA package (Ritchie *et al*, 2015) in R 3.3.1 (R Core Team, 2016). Volcano plots were generated with R base graphics, plotting the non-adjusted *P*-values versus the log2 fold-change and marking data points below 5% FDR (i.e., adjusted *P*-values, Benjamini–Hochberg) when present.

### Data availability

The raw data, MaxQuant output files, and annotated MS2 spectra for all acetylated peptides have been deposited to the ProteomeXchange Consortium (http://proteomecentral.proteomexchange.org) via the PRIDE partner repository with the dataset identifiers PXD006651, PXD006652, PXD006695, PXD006696.

Expanded View for this article is available online.

### Acknowledgements
We would like to thank Jon Nield (Queen Mary, University of London) for the light-reaction figure template, Julia Kimmel for cloning of 35S:HDA14-GFP constructs, and Anne Harzen and Gintaute Dailydaite (MPIPZ) as well as Anne Orwat (LMU Munich) for technical assistance. This work was supported by the Deutsche Forschungsgemeinschaft, Germany (Emmy Noether Programme FI-1655/1-1), and the Max Planck Gesellschaft.

### Author contributions
MF, MH, IF, AB, MP, KK, MS, PJB, J-OJ, and JS performed research; JC performed computational analysis; MH, MS, and IF designed research; DS, DL, GU, GBGM, and MM provided reagents and analytical equipment; MF, MH, J-OJ, KK, and IF analyzed data; IF drafted the manuscript with input from MH, KK, DL, DS, and MES.

### Conflict of interest
The authors declare that they have no conflict of interest.

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
