## [Review Process File · Molecular Systems Biology]

Lysine acetylome profiling uncovers novel histone deacetylase substrate proteins in Arabidopsis

Markus Hartl, Magdalena Füßl, Paul J. Boersema, Jan-Oliver Jost, Katharina Kramer, Ahmet Bakirbas, Julia Sindlinger, Magdalena Plöchinger, Dario Leister, Glen Uhrig, Greg B. G. Moorhead, Jürgen Cox, Michael E. Salvucci, Dirk Schwarzer, Matthias Mann & Iris Finkemeier

Corresponding author: Iris Finkemeier, Max Planck Institute for Plant Breeding Research, Ludwig-Maximilians-University Munich & University of Muenster

Review timeline:

Submission date:	16 June 2017
Editorial Decision:	25 July 2017
Revision received:	21 August 2017
Editorial Decision:	21 September 2017
Revision received:	22 September 2017
Accepted:	25 September 2017

Editor: Maria Polychronidou

Transaction Report:

1st Editorial Decision

25 July 2017

Thank you again for submitting your work to Molecular Systems Biology. We have now heard back from three of the four referees who agreed to evaluate your study. Since their recommendations are quite similar, I prefer to make a decision based on these reports and to not delay the process further by waiting for the report of reviewer #1. As you will see below, the reviewers appreciate that the presented datasets are going to be a useful resource for the plant community. They raise however a series of concerns, which we would ask you to address in a revision of the manuscript.

The reviewers' recommendations are rather clear so I think that there is no need to repeat the points listed below. Please do not hesitate to contact me in case you would like to discuss/clarify any of the points listed by the referees.

 REVIEWER REPORTS

Reviewer #2:

Hartl et al. report on the study of lysine acetylome profiling in Arabidopsis. The results are novel and interesting and for the most part the experiments are well designed. However, there are some serious issues that preclude the publication of this article:

- One serious issues often seen in high-throughput PTM analysis is that the algorithm report FDR

<1%, but are not designed to deal with MSMS that are not from peptides or are due to overlapping peptides. I have issues with the quality of some of the matches. It is not possible to review all of the files; However, I reviewed the file 20140218_EXQ1_MaHa_SA_HDA14LLacK_01 and
Page 2: MS/MS of 413.26 m/z most of the intense ions are not annotated (ie are not part of the sequence)

Page 4: MS/MS of 507.31 m/z most of the intense ions are not annotated (ie are not part of the sequence)

Page 6: MS/MS of 306.91 m/z vast majority of ions are not annotated (ie are not part of the sequence)

Page 8, 15, 27, 30, 41, 43, 44, 46, 49.

This means that out of 50 spectra that I looked at, 12 (24%) had issues that are likely due to multiple peptides present in the MSMS. Yes, the ion series that they use to identify the peptides are present. However, many other ions are not explained and in some cases (the ones I highlighted) have much higher abundances than the ions annotated. Hard to believe that those are confident matches. This needs to be cleaned up.

- The publication would be much stronger if the sites identified had been compared to what is already known in the literature for Arabidopsis and for other species. Are the site truly novel or are they known in other species?

Minor comment:

- The authors use the word "target" too loosely. They have no evidence that they are direct target/substrate. I think that this needs to be made clear. It is clear in the discussion section, but not so in the other section.

Reviewer #3:

The submitted manuscript from Hartl et. al. investigated the regulation of acetylation by lysine deacetylases in Arabidopsis thaliana. The authors initially leveraged an unbiased quantitative proteomic approach using acetyl-lysine peptide affinity enrichment and LC-tandem mass spectrometry to quantify changes in acetylation abundance after treating Arabidopsis leaves with two inhibitors of different specificities. Their finding that the majority of hyperacetylated proteins were nuclear- or cytosol-localized proteins was consistent with the known KDAC activities of the RPD3/HDA1 class, and gave confidence to their overall approach. The acetylome data also helped extend the existing knowledge of this family by finding hyperacetylation of plastid-localized proteins. To identify the potential KDAC in the plastid that could regulate these acetylations, the authors used a clever hydroxamate-based probe to affinity isolate RPD3/HDA1 KDACs from whole leaf extracts. Testing these candidates, the authors were able to confirm by immunofluorescence that HDA14 is localized with the plastids and using hda14 knockouts that the acetylation status of specific plastid proteins is increased. This shows that endogenous HDA14 can regulate lysine acetylation levels.

In general, the targets of specific KDACs are poorly studied across most organisms, including plants. And even more so, the regulatory roles of non-histone acetylation are understudied. This study is a nice example of hypothesis-generating proteomic discoveries of posttranslational modifications accelerating the characterization of specific KDAC protein targets. There are no significant deficits in the experimental approaches. There are several important and minor issues for the authors to address, as described in detail below, but they likely can be addressed by textual addition and clarification. Therefore, after these revisions, this manuscript would be suitable for acceptance.

Primary Issues

1. For the initial discovery experiments, the whole proteome data are presented in Figures 3C and D. However, in general, these datasets seem somewhat under-referenced. For instance, do any of the proteins with differential abundance in the whole proteome overlap with differential acetylated proteins? In other words, what fraction of differential acetylK site abundances are due to bona fide stoichiometry differences? Adding additional interpretation of the data in the Results would serve well in this respect.

2. While the experimental studies with Rubisco and RCA were strengths of the manuscript, their explanation in the Results should include some additional details. For a reader not familiar with

plant biology, the authors should provide biochemical relevance to the concepts of total activity, initial activity, and activation state of RuBiCO, and also summarize the known functional relationship between RuBiCO and RCA. This may help the transition from Rubisco to RCA, which seems abrupt.

Minor Issues

General

1. The manuscript does an adequate job of explaining that acetylation plays roles in regulating plant biochemistry and physiology, however, the larger picture of the relevance of acetylation in plants is not as strong. Is it possible for the authors to provide additional commentary sentences on whether/how understanding the role of acetylation in plants could have an impact on global issues, e.g. , socioeconomics, environment, human health, etc?
2. For the general audience, what would be the ultimate impact Page 12, lines 370 - 374. The authors mention several reasons for why certain targets were more or exclusively upregulated only under apicidin but not TSA, which in theory should exhibit a broader inhibition. In addition to different effective K_i , could the authors' data suggest the specificity of apicidin as a RDA3-specific inhibitor is questionable. What is the prior evidence for this inhibitor specificity, and could its specificity be broader than previously appreciated?
3. Was there a reason the authors did not perform site-directed mutagenesis for RuBisCO as was done for RCA?

Experimental

1. In the Methods section, "Protein Extraction, peptide dimethyl...", it would be helpful to indicate how much protein was processed by the modified-FASP and which centrifugal devices were used.
2. Page 20, line 634-635: The methods list three instruments types used. Were specific instruments used for specific experiments 1 - 5? Or for total proteome versus acetylome? This should be clarified.

Results

1. Page 5, line 147: For context, the number of total identified proteins from the whole proteome analysis could be provided in parentheses, e.g. "(N = x protein)".
2. Page 5, line 160: The authors reference "negatively charged amino acids", but refer to glutamine and asparagine. Did the authors intent to say glutamate and aspartate? E and D would match the sequence logos.
3. Table 1: Is the first set of "protein groups" and "peptides" the whole proteome analyses? If so, it would be useful to label the heading above.
4. Table 1: For the whole proteomes, why did the depth of analysis vary significantly from a low of 2384, to a high of 5107?
5. Related to Table 1, a supplemental Venn diagram showing the unique and shared identified proteins in the acetylome (Table 1) versus whole proteome analyses would be helpful.
6. Figure 2: The legend should indicate why proteins have different number of boxes, e.g., GAPDH has 5, TK has 2.
7. Figure 2: In the legend title, the Calvin-Benson Cycle and light reaction labels are reversed.
8. Page 6, line 191: In addition to the Pearson correlations, a graphical representation of the reproducibility of the approach would be useful. For instance, multiscatter plots plotted with the corresponding Pearson correlation coefficients.
9. Page 7, line 196. The authors used the phrase "as expected" presumably to indicate that it was expected that most acetylation sites are increased with inhibitor treatment. However, the wording of the sentence implies that it was expected that the specific number, 136, lysine acetylatiosn sites were upregulated.
10. Figure 5: The panel letters are missing associated with their respective panels.

Reviewer #4:

Summary

Hartl et al described the use of a quantitative mass spec based proteomics approach to identify and quantify acetylated lysine residues on a proteome wide from the reference plant Arabidopsis

thaliana. For this purpose they use an enrichment strategy based on an anti acetyl-Lysine (acK) antibody coupled to beads. AcK-peptides are enriched from tryptic digest made from leave tissue incubated with two different lysine deacetylase (KDAC) inhibitors. GO-term analysis shows acK-peptides are significantly enriched for several cellular compartments, including nucleus and plastids. Differential lysine acetylation is mostly observed in nuclear proteins and a few plastid proteins. To further explore AcK site on plastid proteins the authors use an activity-based probe to enrich for KDACs from leaf and chloroplast-enrich extracts. HDA14 is specifically enriched in chloroplast extracts and the authors show that HDA14-GFP fusions localize to the chloroplasts and mitochondria. Next they compare the Lysine acetylome of the *hda14* mutant to wild type and find mainly plastid proteins to be differentially acetylated. Finally the effect of lysine acetylation RuBisCO activity and activation state is compared and linked to lysine acetylation K438 of the RCA β 1-isoform, a regulator of Calvin Benson cycle.

With this they make a strong case for a role of lysine acetylation in chloroplast and regulation of the activity and activation state of ribulose-1,5-bisphosphate carboxylase/oxygenase, the key enzyme in photosynthetic CO₂ fixation.

These findings are novel and expand the role of lysine acetylation beyond its well known role in the nuclear histone acetylation.

General remarks.

The study is technically well executed and the proteomics data is made available for inspection through Tables EV1 to 5 as well as through the PRIDE database. The manuscript is well written and the main conclusions are supported by the data. This study is a follow up on a previous study of the plant acetylome by Finkemeier et al (2011). The dataset will be a valuable resource to the plant community and this study a welcome contribution to our understanding of contribution of PTMs in the regulation of response to changes in the environment. I have a few major and minor remarks which are detailed below.

Major remarks:

In the initial dataset generated upon inhibitor treatment the vast majority of the differential acK sites are nuclear, even at lower uncorrected p-values, while the largest proportion of the Lysine acetylated proteins as shown in figure 1C are plastid proteins. Are the inhibitor treatments not effective for plastid localized deacetylases? Only three plastid proteins have differential acK sites after apicidin treatment while TSA treatment resulted in the differential acetylation of only one plastid protein. Based on this conclusion that "The KDAC inhibitor study revealed that most of the RPD3/HDA1 class KDACs of Arabidopsis have their target proteins in the nucleus, but that some members also seem to have their targets in other subcellular compartments, especially in the plastids" is quite a strong statement with an emphasis on plastids that is not really justified based on the numbers of differential acK sites over the total number of acK sites in plastid proteins.

Localization of HDA14 is shown using a 35S-HDA14:GFP construct which seems to be localized in chloroplast and mitochondria. Without any negative control it is difficult to distinguish true plastid localization from an artefact created by using a strong promoter such as 35S. Authors could show a negative control, such as HDA14:GFP lacking the signal peptide driven from the same promoter, as well as use the signal peptide to direct the localization of another fluorescent (fusion) protein.

While the effect of the *hda14* mutation clearly shows differential acK sites, the inhibitor studies show very few differential acK sites. The authors should discuss this apparent discrepancy.

Minor remark:

Mistake in fig2 legend, Calvin Benson cycle shown in fig 2B (not 2A as indicated) and light reactions shown in A (not 2B as indicated)

Reviewer #2:

Hartl et al. report on the study of lysine acetylation profiling in Arabidopsis. The results are novel and interesting and for the most part the experiments are well designed. However, there are some serious issues that preclude the publication of this article:

- One serious issue often seen in high-throughput PTM analysis is that the algorithm reports FDR <1%, but are not designed to deal with MSMS that are not from peptides or are due to overlapping peptides. I have issues with the quality of some of the matches. It is not possible to review all of the files; However, I reviewed the file 20140218_EXQ1_MaHa_SA_HDA14LLacK_01 and

Page 2: MS/MS of 413.26 m/z most of the intense ions are not annotated (ie are not part of the sequence)

Page 4: MS/MS of 507.31 m/z most of the intense ions are not annotated (ie are not part of the sequence)

Page 6: MS/MS of 306.91 m/z vast majority of ions are not annotated (ie are not part of the sequence)

Page 8, 15, 27, 30, 41, 43, 44, 46, 49.

This means that out of 50 spectra that I looked at, 12 (24%) had issues that are likely due to multiple peptides present in the MSMS. Yes, the ion series that they use to identify the peptides are present. However, many other ions are not explained and in some cases (the ones I highlighted) have much higher abundances than the ions annotated. Hard to believe that those are confident matches. This needs to be cleaned up.

We would like to thank the reviewer for looking at the raw data. We agree that the spectra mentioned by the reviewer show abundant ions that are not annotated. However, it has to be noted that the annotations show only a, b, and y ions and neutral losses of NH₃ and H₂O, which corresponds to the fragments used for scoring. There are many additional ions like the precursor, immonium ions, internal fragments, and other less frequently observed neutral losses, which are not annotated but they actually belong to the fragment spectrum. Hence, we do not share the concern that the matches are not confident. Despite our best efforts (see attachment), we were not able to identify an unambiguous overlapping (partial) peptide fragment series. In most cases, the highly abundant fragments in the higher m/z range correspond to the precursor ion and several highly abundant fragments in the lower m/z range correspond to immonium ions. Several spectra show a highly abundant a1 ion, an observation typical of dimethyl labeled peptides. While one example shows fragments that must originate from an overlapping species, the features are clearly distinguishable on MS1 level. Hence, part of the identified MS/MS spectra might be chimeric, in the sense that multiple peptide species have been fragmented in one MS/MS spectrum. However, this is a very common situation in shotgun proteomics and does not pose a fundamental problem. In particular, the MaxQuant/Andromeda workflow is explicitly taking care of this by identifying multiple peptides from one MS/MS spectrum ('second peptide' feature in MaxQuant). FDR calculations are not affected by this. Co-fragmented or chimeric spectra only pose a problem for isobaric labeling techniques, which rely on purity of ion selection for fragmentation, unless more sophisticated methods like MS3-based quantification are used. Here, however we use quantification on MS1 level, which is not affected by co-fragmentation since the co-fragmenting peptides are usually separable on distinguishable MS1 features in the ion isolation window.

In conclusion, close manual inspection of all spectra mentioned by the reviewer did not lead us to dismiss any of those identifications as a (potential) false positive. Therefore, we must strongly disagree with the suggestion of the reviewer that 24% of the spectra have issues.

Finally, we would like to point out that if co-fragmentation was a problem of high-throughput proteomics that cannot be addressed computationally, techniques like DIA, SWATH etc. would not exist – co-fragmentation is the central idea behind them after all.

- The publication would be much stronger if the sites identified had been compared to what is already known in the literature for Arabidopsis and for other species. Are the sites truly novel or are they known in other species?

We agree that this would certainly be an interesting comparison but to our knowledge, there are currently eight acetylomes of higher plants published, with very different depths and degree of annotation and data availability. To perform a thorough comparison of the datasets is beyond the scope of this manuscript and would require a considerable effort of datamining and sequence alignments and thus in our opinion rather be suited for a separate manuscript. However, we have added a comparison of the acetylated proteins and sites for Arabidopsis, which have been described in the literature so far (see p. 6, L175ff). In our manuscript, we reported 959 novel acetylated proteins and 2074 acetylation sites.

Minor comment:

- The authors use the word "target" too loosely. They have no evidence that they are direct target/substrate. I think that this needs to be made clear. It is clear in the discussion section, but not so in the other section.

We have added the words “potential, putative or candidate” in conjunction with KDAC targets or substrates to make this point clearer.

Reviewer #3:

The submitted manuscript from Hartl et. al. investigated the regulation of acetylation by lysine deacetylases in *Arabidopsis thaliana*. The authors initially leveraged an unbiased quantitative proteomic approach using acetyl-lysine peptide affinity enrichment and LC-tandem mass spectrometry to quantify changes in acetylation abundance after treating *Arabidopsis* leaves with two inhibitors of different specificities. Their finding that the majority of hyperacetylated proteins were nuclear- or cytosol-localized proteins was consistent with the known KDAC activities of the RPD3/HDA1 class, and gave confidence to their overall approach. The acetylome data also helped extend the existing knowledge of this family by finding hyperacetylation of plastid-localized proteins. To identify the potential KDAC in the plastid that could regulate these acetylations, the authors used a clever hydroxamate-based probe to affinity isolate RPD3/HDA1 KDACs from whole leaf extracts. Testing these candidates, the authors were able to confirm by immunofluorescence that HDA14 is localized with the plastids and using *hda14* knockouts that the acetylation status of specific plastid proteins is increased. This shows that endogenous HDA14 can regulate lysine acetylation levels.

In general, the targets of specific KDACs are poorly studied across most organisms, including plants. And even more so, the regulatory roles of non-histone acetylation are understudied. This study is a nice example of hypothesis-generating proteomic discoveries of posttranslational modifications accelerating the characterization of specific KDAC protein targets. There are no significant deficits in the experimental approaches. There are several important and minor issues for the authors to address, as described in detail below, but they likely can be addressed by textual addition and clarification. Therefore, after these revisions, this manuscript would be suitable for acceptance.

Primary Issues

1. For the initial discovery experiments, the whole proteome data are presented in Figures 3C and D. However, in general, these datasets seem somewhat under-referenced. For instance, do any of the proteins with differential abundance in the whole proteome overlap with differential acetylated proteins? In other words, what fraction of differential acetylK site abundances are due to bona fide stoichiometry differences? Adding additional interpretation of the data in the Results would serve well in this respect.

We agree and we have added the following paragraph (Page 8, L227ff):

“No significant changes in the regulation of protein abundances were observed after the inhibitor treatments, which covered about 67-88% of proteins carrying the identified acetylated sites (Appendix Fig. S2). However, the whole proteome analysis did not cover very low abundant proteins without enrichment. Therefore, we cannot exclude that the other sites, for which we were not able to quantify protein ratios, were not regulated due to bona fide stoichiometry differences from inhibited KDAC activity. However, we restricted inhibitor treatment to 4h incubation time in order to minimize potential changes in protein abundances that might result from KDAC-dependent alterations in gene expression.”

2. While the experimental studies with Rubisco and RCA were strengths of the manuscript, their explanation in the Results should include some additional details. For a reader not familiar with plant biology, the authors should provide biochemical relevance to the concepts of total activity, initial activity, and activation state of RuBiCO, and also summarize the known functional relationship between RuBiCO and RCA. This may help the transition from Rubisco to RCA, which seems abrupt.

We thank the reviewer for pointing this out. We have added according explanation to make the context more accessible to the readers (see p. 11, 354ff).

“RuBisCO catalyzes the carboxylation or alternatively oxygenation of ribulose-1,5-bisphosphate as the first step in the Calvin-Benson cycle, and thereby enables the photoautotrophic lifestyle of plants. The RuBisCO enzyme activity underlies sophisticated regulation mechanisms, which are dependent on pH, removal of inhibitors, Mg²⁺-ions, and carbamylation of the active site (Portis et al., 2008). Most of these RuBisCO activation steps are dependent on RCA, which is a triple AAA⁺-ATPase enzyme, and which is composed of redox-active alpha isoforms as well as redox-inactive beta-isoforms in Arabidopsis (Carmo-Silva & Salvucci, 2013). The RCA activity itself is inhibited by rising ADP concentrations and remains inactive until the photosynthetic electron transport chain again raises the ATP/ADP-ratio during sunrise. The RuBisCO activation state as well as total activity can be measured by rapid leaf protein extractions (Carmo-Silva et al., 2012).”

Minor Issues

General

1. The manuscript does an adequate job of explaining that acetylation plays roles in regulating plant biochemistry and physiology, however, the larger picture of the relevance of acetylation in plants is not as strong. Is it possible for the authors to provide additional commentary sentences on whether/how understanding the role of acetylation in plants could have an impact on global issues, e.g. , socioeconomics, environment, human health, etc?

We have added the following sentences to the conclusions to make the potential of modulating lysine acetylation in plants more clear (p16, L518ff):

“Since lysine acetylation sites can act as molecular switches, they could be engineered in plant proteins to regulate cell signaling cascades, the expression of certain genes, or to modulate the activities of metabolic enzymes. Furthermore, due to recent advances in advances in CRISPR/CAS technologies, lysine acetylation sites can be used for site-directed mutagenesis also in crop plants. Modifying these lysine residues to constitute acetylated or non-acetylated mimics, ideally will allow a switching of metabolic activities and outputs that have the potential to enhance plant yields or direct metabolism in a way to enhance the accumulation of metabolic intermediates to increase the nutritional values of crops and thereby indirectly promote human health.”

2. For the general audience, what would be the ultimate impact Page 12, lines 370 - 374.

We have added the following sentence for further explanation (Page 14, L435ff): “For example, H3K9 acetylation was found to be associated with actively transcribed genes and has a strong impact on various developmental processes in plants (e.g. Ausin et al., 2004, Benhamed et al., 2006).”

The authors mention several reasons for why certain targets were more or exclusively upregulated only under apicidin but not TSA, which in theory should exhibit a broader inhibition. In addition to different effective K_i , could the authors' data suggest the specificity of apicidin as a RDA3-specific inhibitor is questionable. What is the prior evidence for this inhibitor specificity, and could its specificity be broader than previously appreciated?

To discuss this issue further, we have added the following sentence (Page 13,L407ff):

“Although we cannot exclude that apicidin and TSA have different specificities for Arabidopsis KDACs compared to humans, we observed that recombinant Arabidopsis HDA14, which is a HDA1-like KDAC, is not efficiently inhibited by apicidin but by TSA. This supports the notion that similar specificities of both inhibitors exist for the Arabidopsis KDACs as well.”

3. Was there a reason the authors did not perform site-directed mutagenesis for RuBisCO as was done for RCA?

RuBisCO is a 520 kDa enzyme complex, which consist of eight large and eight small subunits and requires several plant-specific chaperones for assembly (Saschenbrecker et al, 2007, Cell). The assembly of the holo-enzyme is therefore unfortunately not possible in E.coli.

Experimental

1. In the Methods section, "Protein Extraction, peptide dimethyl....", it would be helpful to indicate how much protein was processed by the modified-FASP and which centrifugal devices were used.

We have added the information to Materials and Methods (p. 22, L689).

2. Page 20, line 634-635: The methods list three instruments types used. Were specific instruments used for specific experiments 1 - 5? Or for total proteome versus acetylome? This should be clarified.

The choice of instruments basically depended on availability of instruments over the course of the project. Initially we had compared a Q Exactive instrument to an Orbitrap Elite, measuring the same samples on both machines. We could not identify and performance differences and thus all following samples were measured on Q Exactive type instruments. The information which instruments were used is provided with the raw data in the Pride data depository.

Results

1. Page 5, line 147: For context, the number of total identified proteins from the whole proteome analysis could be provided in parentheses, e.g. "(N = x protein)".

We have added the information.

2. Page 5, line 160: The authors reference "negatively charged amino acids", but refer to glutamine and asparagine. Did the authors intent to say glutamate and aspartate? E and D would match the sequence logos.

We thank the reviewer for spotting this mistake. We have corrected the sentence.

3. Table 1: Is the first set of "protein groups" and "peptides" the whole proteome analyses? If so, it would be useful to label the heading above.

We have added the heading to Table 1.

4. Table 1: For the whole proteomes, why did the depth of analysis vary significantly from a low of 2384, to a high of 5107?

Over the course of the project, we extended the number of fractions for the whole proteome samples to improve coverage of the quantified protein groups, which are lysine-acetylated.

However, fraction only marginally increased the percentages of overlapping quantified acetylation sites and protein groups (see Appendix Fig. S2).

5. Related to Table 1, a supplemental Venn diagram showing the unique and shared identified proteins in the acetylome (Table 1) versus whole proteome analyses would be helpful.

The Venn Diagrams are now added (Appendix Fig. S2)

6. Figure 2: The legend should indicate why proteins have different number of boxes, e.g., GAPDH has 5, TK has 2.

We have added the following sentence to the figure legend and added a new supplementary dataset for linking the boxes to the Arabidopsis identifiers: “For the Calvin-Benson cycle, each box indicates a separate Arabidopsis AGI identifier as indicated in Dataset EV6.”

7. Figure 2: In the legend title, the Calvin-Benson Cycle and light reaction labels are reversed.

Thank you for spotting this mistake. We have corrected the figure legend.

8. Page 6, line 191: In addition to the Pearson correlations, a graphical representation of the reproducibility of the approach would be useful. For instance, multiscatter plots plotted with the corresponding Pearson correlation coefficients.

We have added the scatter plots to the Appendix Suppl. FigS1.

9. Page 7, line 196. The authors used the phrase "as expected" presumably to indicate that it was expected that most acetylation sites are increased with inhibitor treatment. However, the wording of the sentence implies that it was expected that the specific number, 136, lysine acetylation sites were upregulated.

Thank you, we have modified the sentence accordingly (p8, L237)

“As expected for a KDAC inhibitor treatment, most of the lysine acetylation sites (136 in total) were up-regulated (\log_2 -FC 0.4 – 7.4) after apicidin treatment.”

10. Figure 5: The panel letters are missing associated with their respective panels.

Thank you for spotting this. We have added the panel letters to the figure.

Reviewer #4:

Summary

Hartl et al described the use of a quantitative mass spec based proteomics approach to identify and quantify acetylated lysine residues on a proteome wide from the reference plant *Arabidopsis thaliana*. For this purpose they use an enrichment strategy based on an anti acetyl-Lysine (acK) antibody coupled to beads. AcK-peptides are enriched from tryptic digest made from leaf tissue incubated with two different lysine deacetylase (KDAC) inhibitors. GO-term analysis shows acK-peptides are significantly enriched for several cellular compartments, including nucleus and plastids. Differential lysine acetylation is mostly observed in nuclear proteins and a few plastid proteins. To further explore AcK site on plastid proteins the authors use an activity-based probe to enrich for KDACs from leaf and chloroplast-enrich extracts. HDA14 is specifically enriched in chloroplast extracts and the authors show that HDA14-GFP fusions localize to the chloroplasts and mitochondria. Next they compare the Lysine acetylome of the *hda14* mutant to wild type and find mainly plastid proteins to be differentially acetylated. Finally the effect of lysine acetylation RuBisCO activity and activation state is compared and linked to lysine acetylation K438 of the RCA β 1-isoform, a regulator of Calvin Benson cycle.

With this they make a strong case for a role of lysine acetylation in chloroplast and regulation of the activity and activation state of ribulose-1,5-bisphosphate carboxylase/oxygenase, the key enzyme in photosynthetic CO₂ fixation.

These findings are novel and expand the role of lysine acetylation beyond its well known role in the nuclear histone acetylation.

General remarks.

The study is technically well executed and the proteomics data is made available for inspection through Tables EV1 to 5 as well as through the PRIDE database. The manuscript is well written and the main conclusions are supported by the data. This study is a follow up on a previous study of the plant acetylome by Finkemeier et al (2011). The dataset will be a valuable resource to the plant community and this study a welcome contribution to our understanding of contribution of PTMs in the regulation of response to changes in the environment. I have a few major and minor remarks which are detailed below.

Major remarks:

In the initial dataset generated upon inhibitor treatment the vast majority of the differential acK sites are nuclear, even at lower uncorrected p-values, while the largest proportion of the Lysine acetylated proteins as shown in figure 1C are plastid proteins. Are the inhibitor treatments not effective for plastid localized deacetylases? Only three plastid proteins have differential acK sites after apicidin treatment while TSA treatment resulted in the differential acetylation of only one plastid protein. Based on this conclusion that "The KDAC inhibitor study revealed that most of the RPD3/HDA1 class KDACs of *Arabidopsis* have their target proteins in the nucleus, but that some members also seem to have their targets in other subcellular compartments, especially in the plastids" is quite a strong statement with an emphasis on plastids that is not really justified based on the numbers of differential acK sites over the total number of acK sites in plastid proteins.

Thank you for your comments. We have exchanged the word "especially" for "such as" to tone down the emphasis on plastids. Although most of the affected proteins from inhibitor

treatments were nuclear-localized, some were not, and we have followed up on the plastid-localized deacetylase. This is why we have emphasized the plastid in this sentence initially. We added the following sentence to the discussion (Page 14, L441ff):

“Furthermore, our data indicates that TSA might not be effectively taken up into plastids, since the recombinant HDA14 protein was strongly inhibited by TSA, but only few plastid proteins were affected by TSA treatment of Arabidopsis leaves.”

Localization of HDA14 is shown using a 35S-HDA14:GFP construct which seems to localized in chloroplast and mitochondria. Without any negative control it is difficult to distinguish true plastic localization from an artefact created by using a strong promoter such as 35S. Authors could show a negative control, such as HDA14:GFP lacking the signal peptide driven from the same promoter, as well as use the signal peptide to direct the localization of another fluorescent (fusion) protein.

We presented four different types of evidence that HDA14 is localized in the plastids: bioinformatics prediction of signal sequence, GFP-fusion, Asu-pull-down from isolated chloroplasts where only HDA14 but no other HDAs were pulled-down and an indirect evidence in that sense that plastid-encoded proteins are increased in their acetylation status when HDA14 is missing. In a previous publication from our co-authors (Tran et al., 2012, Plant J) it was already shown that N-terminal GFP fusion to HDA14 keeps the protein in the cytosol, as suggested as negative control by the reviewer. Therefore, we believe that already enough evidence was demonstrated that HDA14 is indeed localized in the plastids. However, we have added an additional experiment of a Western-blot analysis with isolated plastids and mitochondria from WT, *hda14* and HDA14-GFP lines showing that the protein is residing within the chloroplast stroma and in mitochondria (Appendix FigS4).

While the effect of the *hda14* mutation clearly shows differential acK sites, the inhibitor studies show very few differential acK sites. The authors should discuss this apparent discrepancy.

We have added the following sentence: Page 14, Line 441ff

“Furthermore, our data indicates that TSA might not be effectively taken up into plastids, since the recombinant HDA14 protein was strongly inhibited by TSA, but only few plastid proteins were affected by TSA treatment of Arabidopsis leaves.”

Minor remark:

Mistake in fig2 legend, Calvin benson cycle shown in fig 2B (not 2A as indicated) and light reactions shown in A (not 2B as indicated)

Thank you for spotting this. We have corrected the figure legend.

Reviewer #1:

Summary

Acetylation of lysine residues in proteins has long been seen as a hallmark for epigenetic control of histones. More recently it became evident that a whole acetylome of non-histone like proteins exists in major subcellular compartments that is reversible by the action of lysine acetylases and de-acetylases (KDACS) and can affect properties of proteins such as enzymatic activities. Here a comprehensive inventory of lysine acetylation sites is reported with emphasis on targets of one subtype of KDACS, chloroplast proteins, a novel deacetylase and two selected functional examples.

The authors provide an in-depth acetylome site analysis of proteins from *Arabidopsis* using a state-of-the-art isotope demethylation labelling strategy and immune-affinity enrichment by anti-acetyl-lysine agarose beads. Both techniques had been established previously. This yielded 20 times more lysine acetylation sites than known before from *Arabidopsis*. More than 40% of these proteins were attributed to chloroplasts, possibly not surprising because these are most abundant in green leaves. The study focuses on the RPD/HDA1 subtype of KDACS using two specific inhibitors to determine novel acetylation sites of this type. To explain the high proportion of target sites in plastids, three KDACS of the RPD/HDA1 type were affinity purified and the enzymatic activity and plastid localization of one of them, HDA14, demonstrated. Contrary to earlier findings (Tran et al., 2012) the here described HDA14 protein is neither found to be a histone deacetylase nor localized to cytosol and nucleus. Consequently, a *hda14* knock-out/down mutant was analysed and considerable differences in the abundance of acetylated sites of chloroplast and thylakoid proteins compared to wild type were observed. Out of these, 14 target proteins are encoded in the plastome, underpinning the relevance of this process in the organelle. To test the physiological relevance of these acetylations, *hda14* mutant plants were transferred from normal to low light conditions. In comparison to wildtype, 32 protein groups, mostly related to photosynthesis, showed altered abundances of acetylated sites at unchanged protein levels, pointing to acetylation as possible regulatory process. Among those proteins Rubisco activase (RCA) is strongly affected, which correlates with elevated Rubisco activity and activation state in *hda14* mutants. When the authors mutated a major acetylation site in RCA subunit beta1 this resulted in a lowered inhibition by ADP, a known factor of RCA regulation. The major conclusion from this is that lysine acetylation at the K438 site causes higher RCA activity and therefore Rubisco activity under low light growth conditions.

Taken together, the concept aims at drawing a line from changes at large scale post-translational lysine acetylation patterns to regulation of the central enzyme of photosynthesis as evidence for the physiological importance of lysine acetylation of non-histone

proteins.

General remarks

The work represents a significant advance with respect to lysine acetylation sites, making clear that this is a massive and so far often underestimated post-translational modification. As far as the reviewer can judge, the work is of very high quality in terms of protein chemistry and will provide an excellent and fruitful of novel lysine acetylation sites for further research. It exceeds previous studies of this kind, at least in plants, not only by quality and comprehensiveness, but also by the attempt to provide functional evidence for altered acetylation patterns in general and specifically in proteins in response to environmental change. These are clear advances that will make this study interesting for the acetylation field in all eukaryotic models and the plant field in particular. However, the functional links presented here show some weaknesses. This firstly refers to the apparent lack of a physiological phenotype of the *hda14* mutant used as a genetic approach to show relevance of lysine acetylation. While acetylation patterns change differently compared to wildtype after transfer to low light, the biological meaning of these changes remains unclear. Another point is the lack investigation of other family members of KDACs whose up-regulation might affect the acetylation patterns, especially given the limited knowledge on substrate specificities.

Major criticisms:

-Despite the wealth of data it not easy to find out the number of analysed proteins, the efficiency of enrichment and the number of proteins (not protein groups) that are lysine acetylated.

The number of analysed proteins is presented in Table 1, listed by experiment and as the total number across experiments. In our opinion, it is more meaningful and honest to differentiate and count protein groups rather than individual proteins, as there is no experimental evidence that allows telling which of the proteins in a particular group were actually present in the sample.

-The substantial number of identified sites allows for predictions of acetylation sites. It would be very interesting to endeavor a prediction of potential sites in the Arabidopsis proteome - with necessary caution of course.

We agree that it would certainly be very interesting to use this dataset to train algorithms for proteome-wide prediction of sites. However, this is beyond the scope of the manuscript and should rather be treated in a separate paper. We present iceLogos for the different subcellular compartments (Fig. 1), which shall be useful for researcher to look for putative acetylation sites in their proteins of interest.

-Why are nuclear and plastid proteins over-represented as acetylation targets and ER, vacuole, mitochondria, plasma membrane and extracellular space proteins underrepresented?

There are at least two possible explanations for this observation. First, more abundant proteins (such as nuclear or plastidial proteins) are more likely to be sampled than less abundant

proteins (e.g. mitochondrial proteins). Although our fractionation and enrichment scheme allowed deep coverage, it is certainly possible that low abundance proteins/sites are still under-represented. Second, lysine acetylation might preferentially occur in different subcellular compartments in different tissues or developmental stages, which might point to its role in particular physiological processes and could also reflect evolutionary history. At least in *Arabidopsis* seedlings many mitochondrial proteins can be found acetylated (König et al., 2014, Mitochondrion). Furthermore, the plastid/nucleus overrepresentation, which we found for *Arabidopsis* leaves, actually fits to the predicted localization of most of the plant acetyltransferases (Uhrig et al., 2017 BMC Genomics).

-HDA14 enzyme activity assay: are there controls w/o metals? Any time dependency of the reaction? It seems the most simple assay tests for an enzyme are not shown.

We have now added the description to materials and methods how the apoenzyme was prepared to make this point clearer. The enzyme is not active without the metal co-factor. The data presented show the turnover of substrate per second. This is a very straightforward enzyme assay, which was used in previous publications from our co-authors on human HDACs before (Dose et al., 2012 Chem Comm).

-What is the phenotype of the *hda14* mutant (other than altered acetylation patterns), possibly during the different light treatments? This would be essential to allow to decide if lysine acetylation has a relevant biological function or is merely a reflection of general acclimation processes, e.g. acetyl-coenzyme A status. Are there expression changes of the other KDAC encoding genes in *hda14*? Is the *hda14* line a knock-down (mRNA left) or knock-out (no protein detectable)? A second allele would be advisable at least for some experiments.

The *hda14* mutant line is a T-DNA insertion mutant and as such a full KO with no protein detectable as shown in Appendix FigS6. The residual transcript detected in the knock-out results from a truncated transcript. Under standard conditions in a growth chamber *hda14* does not show an obvious growth phenotype. We have now added the fresh weight measurements to the Appendix FigS6. It is a common problem in (plant) research that many gene knock-outs especially from regulatory proteins do not display obvious growth or fitness phenotypes under standard growth conditions. Research groups specialized in molecular ecology have demonstrated that plants, which are perfectly fine under standard conditions, reveal severe phenotypes only when grown under field conditions, where the plants are simultaneously exposed to an array of environmental challenges. Consequently, it would be possible that the *hda14* phenotype becomes evident only under specific growth conditions, which are difficult to determine in a growth chamber.

We are convinced that the differential acetylation patterns in *hda14* under different light conditions and the follow-up analysis with mutated RCA provide sufficient evidence for a functional role of HDA14 in photosynthetic regulation.

-If there is a phenotype, it should be exploited to demonstrate the biological significance of K438 mutations in the RCA subunit using transformation of either wildtype or *hda14* mutants. Can the phenotype be rescued? Can the phenotype be induced or phenocopied by directed mutation using e.g. Crspr or complementation of the null mutant? This work presents strong correlations, but no sufficiently proven evidence.

See comments above.

-Why is more RCA acetylation and Rubisco activity observed after transfer to low light? One would expect (and there is literature for this) that the activity decreases. This would make sense due to the coupling to the light reactions but also to lowered acetyl-coenzyme A levels under low light.

ADP usually inhibits the RCA activity under low light conditions. However, our acetylation mimic mutants show that the ADP inhibition of RCA is much less pronounced, when the enzyme is acetylated at K438. In WT plants, RCA-K438 is under the control of HDA14, which keeps the enzyme in its deacetylated form, and hence, RCA is ADP-sensitive and more inhibited in the WT under low light. In the *hda14* mutant, the lack of HDA14 cannot prevent acetylation of RCA-K438, and thus RCA is more active and a higher Rubisco activation state is observed. This is also outlined in the discussion, p16 L489ff.

Conclusion: A comprehensive study of high quality at the proteomic side but weaknesses at the functional side. To exceed an excellent inventory of lysine acetylation sites, the proof that changes of acetylation patterns of specific target proteins is of biological relevance needs to be provided.

We thank the reviewer for the thorough analysis and the helpful comments, which allowed us to further improve the manuscript. This is a systems analysis on histone deacetylase target proteins and with regard to the conclusion, we disagree that further proof needs to be provided on this level. It would certainly be desirable to present a full-blown story that pins down the physiological function of HDA14 in all detail, but this lies beyond the scope of this manuscript. In this manuscript, we are presenting a large-scale technically well-conducted study (as the reviewer points out himself), which in itself represents a considerable effort and took already more than five years to set-up and conduct, which is also reflected by the many co-authors involved in this study. We followed this up with a series of experiments in which we provide additional evidence for the role of lysine deacetylation in chloroplasts and photosynthesis, by identifying the key KDAC enzyme in chloroplasts and its potential target sites, and by showing how one of these sites might act as a regulatory site for a key photosynthetic process. In the current revised state, we are convinced that the manuscript provides a large set of interesting and useful data and of sufficiently justified observations and conclusions that are of wide importance for the field. Trying to follow-up on the functional implications in all detail, as suggested by the reviewer, would probably require several years of additional research with unknown outcome and delay the provision of this study.

Attachment: Inspection of MS2 spectra (for reviewer 2)

Scan	Page # Original	Page # This file	Comment
10408	2	2	Further automated annotation explains high intensity signals
11020	4	3	Further automated annotation explains majority of high intensity signals
11883	6	4	Manual annotation explains several high intensity signals; distance between signals in the higher m/z range does not correspond to amino acids with only one exception
12388	8	5	Manual annotation does not identify a clear and unambiguous overlapping sequence although several mass differences between signals correspond to amino acid masses
13071	15	6	Further automated annotation explains several high intensity signals in the lower m/z range; peak distances in the higher m/z range do not correspond to amino acids with only one exception
15665	27	7	Further automated annotation explains high intensity signals
16877	30	8	Further automated annotation explains several high intensity signals; manual annotation does not reveal any peak distances corresponding to amino acid masses
19150	41	9	Manual annotation finds several peak distances that do not correspond to amino acid masses; distances between m/z 1421 and either 1263 or 1197 could correspond to amino acid combinations, altogether not enough evidence for co-fragmenting peptide
20287	43	10	Further automated annotation explains high intensity signals
20393	44	11, 12	Signals beyond precursor suggest co-fragmenting species of higher mass; MS1 scan shows that signals of peptide of interest at 591.8596, 592.3594, 592.8609 are baseline separated from other signals; no real concern for identification as well as quantification of the peptide of interest
20668	46	13	Only high intensity signal without annotation in original file corresponds to precursor
21246	49	14	High intensity signal without annotation in high m/z range corresponds to precursor

Raw file 20140218_EXQ1_MaHa_SA_HDA14LLacK_01 Scan 10408 Method FTMS; HCD Score 81.35 m/z 413.26

Raw file 20140218_EXQ1_MaHa_SA_HDA14LLacK_01 Scan 10408 Method FTMS; HCD Score 81.35 m/z 413.26

Raw file
20140218_EXQ1_MaHa_SA_HDA14LLack_01

Scan 11020 Method FTMS; HCD Score 113.46 m/z 507.31

Raw file
20140218_EXQ1_MaHa_SA_HDA14LLack_01

Scan 11020 Method FTMS; HCD Score 113.46 m/z 507.31

Raw file
20140218_EXQ1_MaHa_SA_HDA14LLack_01

Scan 11883 Method FTMS; HCD Score 80.79 m/z 306.91

Raw file
20140218_EXQ1_MaHa_SA_HDA14LLack_01

Scan 11883 Method FTMS; HCD Score 80.79 m/z 306.91

Raw file 20140218_EXQ1_MaHa_SA_HDA14LLack_01 Scan 12388 Method FTMS; HCD Score 96.82 m/z 373.74

Raw file 20140218_EXQ1_MaHa_SA_HDA14LLack_01 Scan 12388 Method FTMS; HCD Score 96.82 m/z 373.74

Raw file
20140218_EXQ1_MaHa_SA_HDA14LLack_01

Scan 13071 Method FTMS; HCD Score 133.23 m/z 382.88

Raw file
20140218_EXQ1_MaHa_SA_HDA14LLack_01

Scan 13071 Method FTMS; HCD Score 133.23 m/z 382.88

Raw file
20140218_EXQ1_MaHa_SA_HDA14LLack_01

Scan	Method	Score	m/z
15665	FTMS; HCD	67.93	552.82

Raw file
20140218_EXQ1_MaHa_SA_HDA14LLack_01

Scan	Method	Score	m/z
15665	FTMS; HCD	67.93	552.82

Raw file
20140218_EXQ1_MaHa_SA_HDA14LLacK_01

Scan 16877 Method FTMS; HCD Score 84.48 m/z 454.61

Raw file
20140218_EXQ1_MaHa_SA_HDA14LLacK_01

Scan 16877 Method FTMS; HCD Score 84.48 m/z 454.61

Raw file
20140218_EXQ1_MaHa_SA_HDA14LLack_01

Scan
19150

Method
FTMS; HCD

Score
81.65

m/z
541.33

Raw file
20140218_EXQ1_MaHa_SA_HDA14LLack_01

Scan
19150

Method
FTMS; HCD

Score
81.65

m/z
541.33

Raw file 20140218_EXQ1_MaHa_SA_HDA14LLack_01 Scan 20287 Method FTMS; HCD Score 96.03 m/z 549.83

Raw file 20140218_EXQ1_MaHa_SA_HDA14LLack_01 Scan 20287 Method FTMS; HCD Score 96.03 m/z 549.83

Raw file 20140218_EXQ1_MaHa_SA_HDA14LLack_01 Scan 20393 Method FTMS; HCD Score 72.5 m/z 591.86

Raw file 20140218_EXQ1_MaHa_SA_HDA14LLack_01 Scan 20393 Method FTMS; HCD Score 72.5 m/z 591.86

Raw file
20140218_EXQ1_MaHa_SA_HDA14LLack_01

Scan
20668

Method
FTMS; HCD

Score
142.39

m/z
567.98

Raw file
20140218_EXQ1_MaHa_SA_HDA14LLack_01

Scan
20668

Method
FTMS; HCD

Score
142.39

m/z
567.98

Raw file
20140218_EXQ1_MaHa_SA_HDA14LLacK_01

Scan	Method	Score	m/z
21246	FTMS; HCD	58.6	551.85

Raw file
20140218_EXQ1_MaHa_SA_HDA14LLacK_01

Scan	Method	Score	m/z
21246	FTMS; HCD	58.6	551.85

2nd Editorial Decision

21 September 2017

Thank you again for sending us your revised manuscript. We have now heard back from the referee who was asked to evaluate your study. As you will see below, s/he is satisfied with the modifications made. Therefore we think that the study is now suitable for publication in *Molecular Systems Biology*.

Before we formally accept the study for publication, we would ask you to address some minor editorial issues listed below.

REVIEWER REPORT

Reviewer #2:

The authors have resolved my concerns

Corresponding Author Name: Prof. Dr. Iris Finkemeier

Manuscript Number: MSB-17-7819